# D³: Divide, Describe, and Diffuse: Prompt-Enriched, Scene-Aware Dataset Condensation for Object Detection

## Abstract

Dataset condensation (DC) seeks to compress large datasets into small synthetic ones for efficient training. Recent work applies text-to-image diffusion models to DC, but their naïve use in object detection creates a representation bottleneck: conditioning on short captions yields sparse, single-object scenes with limited spatial coverage, failing to capture the multi-object co-occurrence and layout diversity essential for detector generalization. We present D³ (Divide, Describe, Diffuse), a light-weight framework that produces dense and semantically diverse synthetic images tailored for detection. D³ constructs a scene-to-object dictionary from dataset statistics, which guides a Large Language Model to generate enriched captions grounded in realistic contexts. To further enhance fidelity and spatial variety, we design two structured prompting strategies: prompt merging, which combines multiple enriched captions, and spatial partitioning, which allocates sub-prompts to different image regions. On MS COCO and PASCAL VOC, D³ achieves state-of-the-art performance under severe data constraints. Notably, using only 0.5% of VOC data, D³ reaches 18.8% mAP (vs. 8.5% for the strongest baseline), demonstrating its effectiveness in overcoming the representation gap of prior DC methods.

## 1 Introduction

Dataset Condensation (DC) is an emerging topic nowadays, aiming to generate a small set of condensed images from a larger one to accelerate the training of neural networks. Thanks to its efficiency in accelerating training and reducing storage requirements, DC has found applications in various areas, including continual learning(Yang et al., 2023; Gu et al., 2024b), image super-resolution(Zhang et al., 2024), privacy-preserving (Dong et al., 2022; Chai et al., 2024), medical (Jin et al., 2025; Kanagavelu et al., 2024), and federated learning (Holland et al., 2024; Yan et al., 2025).

Despite substantial progress in dataset condensation for image classification, demonstrated by recent methods (Deng et al., 2024; Son et al., 2024; Wang et al., 2025b; Shin et al., 2025; Wang et al., 2025a), extending these techniques to object detection remains hardly explored. Most dataset condensation methods for classification rely on heavy optimization, where synthetic images and networks are updated jointly through bi-level gradient matching objectives (Zhao et al., 2021). Extending this recipe to detection is computationally prohibitive because training repeatedly back-propagates high-resolution images through large detectors (e.g., FPN/transformer backbones with multi-scale heads), and the

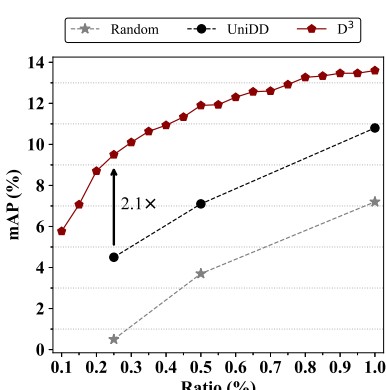

Figure 1: **Performance *vs.* training-data ratio**. On COCO with Faster R-CNN (Ren et al., 2015), under an extreme data constraint (0.25% storage budget), D³ achieves more than double the mAP of UniDD (Qi et al., 2025).

memory/time cost scales with the number of instances per image. In practice, this makes end-to-end distillation for detection slow and costly.

To date, progress in DC for object detection has been limited. One of the earliest attempts, DCOD (Qi et al., 2024), adopted a two-stage strategy: first pretraining a detector on the full dataset, then synthesizing images via model inversion. More recently, text-to-image (T2I) generative models have been explored, as in UniDD (Qi et al., 2025), to directly synthesize training data. While diffusion-based T2I models can produce visually realistic images, their naïve application to detection introduces a fundamental learning bottleneck. The conditioning signal, e.g., short image captions such as "a cat on a sofa", compresses the high-dimensional semantics of detection into a sparse, low-entropy description. As a result, the generated data exhibit low object density, limited spatial coverage, and weak contextual relationships. From a learning perspective, this creates an information bottleneck: the synthetic distribution undersamples multi-object co-occurrence and diverse layouts, which are crucial for learning robust detector representations. Consequently, detectors trained on such data fail to generalize to cluttered, real-world scenes.

To address this challenge, we design a caption enhancement pipeline that enriches both the semantic and structural content of captions before image generation, enabling the synthesis of content-rich, detection-friendly images. At its core lies a scene-to-object dictionary, which bridges scene semantics and object composition. The dictionary is automatically built by clustering training images into scene categories and aggregating frequently co-occurring objects per scene. Unlike handcrafted templates, it is learned directly from dataset statistics, capturing realistic co-occurrence patterns that ground prompts in visual context.

We leverage this dictionary for Scene-Aware Caption Enrichment, where a Large Language Model (LLM) expands base captions with scene-consistent object terms. Enriched captions contain multiple relevant entities, improving both semantic diversity and contextual grounding. To further enhance generation quality, we introduce two composition strategies: caption merging, which combines multiple enriched captions into a single caption to increase semantic richness, and spatial partitioning, which assigns captions to predefined regions of the image canvas, promoting spatial diversity and reducing overlaps. By unifying dataset-driven enrichment with spatially structured captioning, our framework generates synthetic images that preserve multi-object co-occurrence and spatial relations—key factors for robust representation learning in detection.

Our main contributions are summarized as follows:

- A scene-to-object dictionary that grounds LLM-generated prompts in dataset-level co-occurrence statistics, mitigating the information bottleneck of short captions.
- A two-stage caption enhancement pipeline consisting of Scene-Aware Caption Enrichment and Multi-Caption Composition (caption merging + spatial partitioning), enriching both semantic and structural representations.
- A new framework, $D^3$ (Divide, Describe, Diffuse), that achieves state-of-the-art performance under extreme data constraints, more than doubling the strongest baselines on PASCAL VOC and MS COCO at 0.25% of the storage budget, as shown in Figure 1.

## 2 RELATED WORK

**Dataset Condensation for Image Classification.** Dataset condensation (DC) has been extensively studied in the context of image classification, where the goal is to distill a large dataset into a much smaller synthetic one while maintaining competitive performance(Wang et al., 2018; Zhao et al., 2021; Kim et al., 2022). Classification offers a relatively simple setting, as each image typically contains a single object and a single label. A wide range of approaches have been proposed to enhance the fidelity and generalization of condensed datasets, including gradient matching(Zhao et al., 2021), trajectory-based optimization(Cazenavette et al., 2022), and more recent improvements in distribution matching and regularization (Zhao & Bilen, 2022a; Zhao et al., 2023). However, these methods rely heavily on the structural simplicity of classification and are not directly applicable to more complex structured tasks such as object detection.

**Generative Models for Image Classification.** To overcome the scalability limits of pixel-space optimization, recent works have explored generative models for dataset condensation. Instead of di-

rectly optimizing pixels, these methods synthesize images from random noise(Zhao & Bilen, 2022b; Cazenavette et al., 2023; Zhong et al., 2024) or textual descriptions(Gu et al., 2024a; Su et al., 2024; Zhao et al., 2025a; Chan Santiago et al., 2025). By leveraging powerful pretrained generators such as diffusion models or GANs, they adapt to target datasets either by fine-tuning to reduce distribution mismatch(Cazenavette et al., 2023), or by explicitly promoting higher sample diversity(Gu et al., 2024a). These approaches offer a scalable alternative to pixel-space condensation in classification. However, their success still hinges on single-label, single-object supervision, leaving open the challenge of extending generative DC to structured tasks such as object detection.

**Dataset Condensation for Object Detection.** Compared to classification, extending dataset condensation to detection introduces additional challenges, as models must learn representations of multiple objects with diverse spatial configurations. One of the earliest attempts, DCOD(Qi et al., 2024), adopts a two-stage pipeline: in the *forge* stage, a detector is trained on real data, and in the *fetch* stage, condensed images are recon-

Table 1: **Comparison of image generation strategies for object detection condensation.**

| Attribute | DCOD | UniDD | $D^3$ (Ours) |
|---|---|---|---|
| Spatial control | None | Implicit | Explicit |
| Scene enrichment | None | ✗ | ✓(Dictionary) |
| Prompt type | None | Single | Structured |
| Object density | Sparse | Sparse | Dense |

structed via model inversion. While pioneering, this approach initializes synthetic images from randomly selected real samples, resulting in limited coverage of the image space. More recently, UniDD(Qi et al., 2025) proposed a unified generative framework for classification, detection, and segmentation. However, UniDD's images often lack semantic density, failing to capture rich multi-object co-occurrence and balanced class diversity. These limitations highlight the need for methods that explicitly enrich the visual and semantic content of each image, producing condensed datasets that are compact yet dense and diverse—key properties for improving detection performance. Table 1 summarizes the differences between our approach and prior work.

## 3 MOTIVATING EXAMPLE

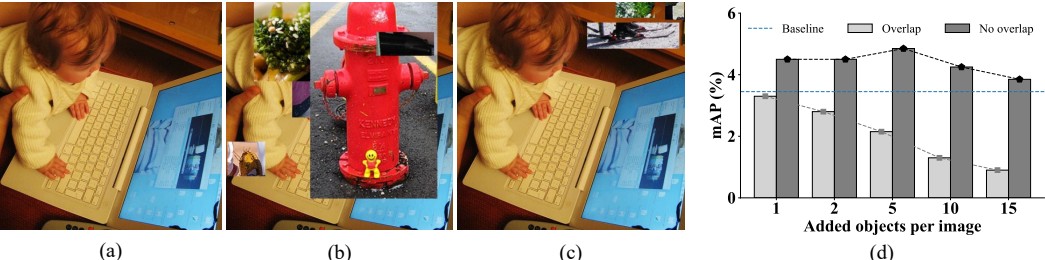

(a)        (b)        (c)        (d)

Figure 2: **Overlap hurts; non-overlap helps modestly.** (a) Original image. (b–c) Copy–Paste with overlapping vs. non-overlapping inserts. (d) Performance on COCO dataset (at 0.25% of data) of Faster R-CNN; the *Baseline* is trained without Copy–Paste.

**Setup.** Under strict data budgets, training images are sparse with few objects and limited co-occurrence so detectors learn weak spatial context. To enrich the training data, we consider a simple pixel-space control (Copy–Paste (Ghiasi et al., 2021) / CutMix (Yun et al., 2019)): crop an instance and paste it into another image. For each target image, we randomly paste $K \in \{1, 2, 5, 10, 15\}$ instances using instance cutouts at random scale. We evaluate two variants: *overlap-allowed* and *no-overlap* (intersection-over-union $= 0$ w.r.t. existing and pasted boxes).

**Results.** As show in Figure 2(d), allowing overlap gradually *decrease* performance as the number of added objects increases. We conjecture that overlap degrades supervision by (i) occluding ground-truth instances while their annotations remain visible-only (amodal–visible mismatch), (ii) inducing awkward truncations and depth/scale inconsistencies that act as noise (see Figure 2 (b)). Meanwhile, enforcing no overlap gives a modest lift (to $4.25$), as empty regions can be filled with additional supervision. However, as more instances are pasted per image, the gains quickly *saturate* and can reverse—naïve insertion increases count but not *scene coherence* (plausible placement, scale, and context). This suggests that improvements are affected not only by instance count but also by the realism of the generated scenes.

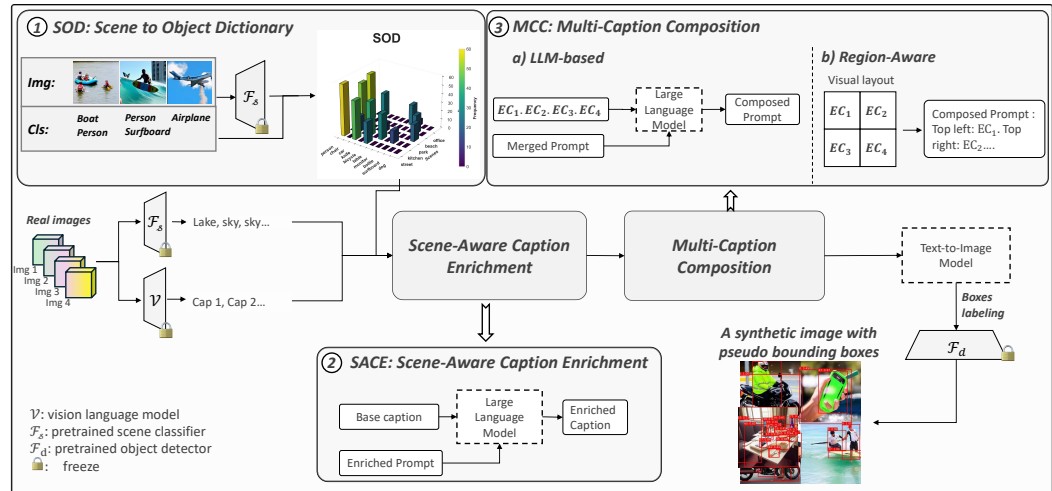

Figure 3: **Overview of our framework.** It consists of three stages: (1) constructing a Scene-to-Object Dictionary (SOD) from dataset co-occurrence statistics, (2) enriching base captions via Scene-Aware Caption Enrichment (SACE), and (3) composing structured prompts through Multi-Caption Composition (MCC) to guide text-to-image generation.

Instead of injecting additional pixels that compromise scene realism, we therefore **augment in text space** and then **realize in image space**: captions are enriched with scene-aware object sets and region cues, and images are synthesized to respect these constraints. This text-to-image route plans *global layout* and *object co-occurrence* before rendering, ensuring consistent occlusions, backgrounds, and lighting. The result is dense, contextually coherent images with clean labels, which in turn drive stronger downstream performance, motivating our $D^3$ pipeline.

## 4 METHOD

### 4.1 IMAGE-TO-TEXT PAIR GENERATION

To guide T2I models in generating synthetic training data, we first extract natural language descriptions from real images in the detection dataset. These descriptions summarize key objects and scene context, serving as baseline prompts.

We employ a pretrained vision-language model $\mathcal{V}$ (e.g., BLIP (Li et al., 2022)) to generate captions. Given an image $x \in \mathbb{R}^{H \times W \times 3}$, the corresponding caption is obtained as $c = \mathcal{V}(x)$, where $c$ is a free-form caption capturing salient visual elements in $x$.

The resulting $(x, c)$ pairs form a caption pool that both anchors our enrichment process to the real data distribution and provides input prompts for subsequent synthesis.

### 4.2 SCENE-AWARE CAPTION ENRICHMENT

Captions extracted from real images often describe only a single object or a coarse scene label, failing to capture the full diversity required for object detection. Using such captions as prompts for T2I generation typically produces synthetic images with low object density and limited semantic structure. To address this, we propose Scene-Aware Caption Enrichment (SACE), which augments captions with additional objects drawn from dataset-level co-occurrence statistics. This grounds prompts in realistic scene–object relationships and encourages the generation of semantically rich multi-object images.

**Scene-to-Object Dictionary Construction.** Object distributions vary significantly across scene types (e.g., microwave, sink, and bottle frequently co-occur in kitchens). We therefore construct a dataset-driven scene-to-object dictionary $\mathcal{D}$ that maps each scene category $s \in \mathcal{S}$ to its top-$K$ most

frequent object classes:

$$\mathcal{D}(s) = \texttt{Top-}K\{\texttt{Objects}(x_i) \mid x_i \in \mathcal{I}_s\},$$

where $\mathcal{I}_s$ is the set of images labeled as scene $s$, and $\texttt{Objects}(x_i)$ returns annotated object labels. A pretrained scene classifier $\mathcal{F}_{\text{scene}}$ (e.g., Places365(Zhou et al., 2017)) assigns each image a scene label $s_i = \mathcal{F}_{\text{scene}}(x_i)$.

**Scene-Guided Caption Synthesis.** At generation time, given an image–caption pair $(x, c)$, we first infer the scene category

$$s = \mathcal{F}_{\text{scene}}(x).$$

We then retrieve the relevant object classes $\mathcal{D}(s)$ and condition a language model $\mathcal{L}$ on both $c$ and $\mathcal{D}(s)$ to produce an enriched caption:

$$c^+ = \mathcal{L}(c \mid \mathcal{D}(s)),$$

where $c^+$ augments the original caption with frequently co-occurring objects consistent with scene context $s$.

As shown in Figure 4, this enrichment strategy encourages the caption to contain dense, realistic multi-object compositions, embedding co-occurrence structures that are critical for robust representation learning in detection.

*Prompt:* Enrich the caption by smoothly incorporating as many contextually appropriate objects from the object list.
*Example:*
<caption>: A set **table** with breakfast foods and empty **glasses**.
<object list>: **sink, fork, clock, knife, couch, cell phone, wine glass, bowl, remote, handbag**

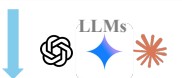
**LLMs**

*Enriched caption:*
A set **table** with breakfast foods, empty **glasses**, **bowls**, **forks**, **knives**, and a **cell phone** sits near a **couch**, with a **clock** hanging on the wall.

Figure 4: **Caption enrichment example.** Newly inserted objects are shown in **bold**; unchanged text is in gray.

### 4.3 MULTI-CAPTION COMPOSITION FOR DENSE GENERATION

While SACE improves object diversity at the semantic level, it does not explicitly control the spatial distribution of objects in the generated image. This often results in object clustering or large empty regions. To address this, we introduce two prompt composition strategies aimed at promoting spatial diversity and dense object layouts: (1) *LLM-based Merging*, and (2) *Region-Aware Composition*.

**LLM-based Merging.** Let $\mathcal{C} = \{c_1, c_2, \ldots, c_N\}$ be a pool of captions. We sample a subset $\mathcal{C}_m = \{c_{i_1}, \ldots, c_{i_m}\} \subset \mathcal{C}$ and use a language model $\mathcal{L}$ to synthesize a unified, multi-object prompt:

$$\hat{c} = \mathcal{L}(c_{i_1}, \ldots, c_{i_m}). \tag{1}$$

This composition introduces object diversity and scene complexity by blending multiple semantically distinct captions into a single prompt. As a result, the generated images are more likely to contain a wider variety of objects, increasing the chances of producing detection-friendly scenes with higher object density.

**Region-Aware Composition.** To guide the T2I model with structural layout cues, we divide the image canvas into $K$ spatial regions (e.g., top-left, top-right, etc.) and assign a distinct caption $c_k$ to each region $R_k$:

$$\{(R_k, c_k)\}_{k=1}^{K}, \quad \text{where } c_k \sim \mathcal{C}. \tag{2}$$

We then compose a structured prompt by concatenating region-aware descriptions:

$$\hat{c} = \texttt{``Top left: } c_1. \quad \texttt{Top right: } c_2. \quad \texttt{...''} \tag{3}$$

This strategy introduces a *soft layout constraint* such that the model is guided to place distinct object groups in specified regions, solely through linguistic prompting. While it does not enforce explicit spatial alignment, this method effectively encourages spatial diversity and dense object layouts. Crucially, it scales scene complexity without requiring any spatial supervision such as bounding boxes or segmentation masks, making it fully compatible with standard T2I pipelines.

Table 2: **Results on VOC**. $D^3$ achieves superior performance across all ratios, especially under extreme data scarcity. The last column indicates the performance gain over UniDD.

| Ratio | Metric | Random | Uniform | K-Center | Herding | DCOD | UniDD | $D^3$ (Ours) | Gain |
|---|---|---|---|---|---|---|---|---|---|
| 0.5% | mAP | $0.8_{\pm0.2}$ | $0.9_{\pm0.1}$ | $0.5_{\pm0.1}$ | $0.6_{\pm0.2}$ | $6.3_{\pm0.4}$ | $8.5_{\pm0.4}$ | $\mathbf{18.8_{\pm0.1}}$ | ↑**10.3** |
| | $AP_{50}$ | $3.1_{\pm0.4}$ | $3.4_{\pm0.3}$ | $2.1_{\pm0.3}$ | $2.4_{\pm0.3}$ | $18.5_{\pm0.7}$ | $22.3_{\pm0.6}$ | $\mathbf{43.6_{\pm0.3}}$ | ↑**21.3** |
| | $AP_{75}$ | – | – | – | – | – | – | $\mathbf{13.0_{\pm0.3}}$ | – |
| 1% | mAP | $4.2_{\pm0.5}$ | $5.7_{\pm0.2}$ | $3.6_{\pm0.6}$ | $3.5_{\pm0.5}$ | $13.8_{\pm0.5}$ | $16.8_{\pm0.5}$ | $\mathbf{22.9_{\pm0.1}}$ | ↑**6.10** |
| | $AP_{50}$ | $13.7_{\pm0.6}$ | $17.7_{\pm0.4}$ | $12.3_{\pm0.3}$ | $11.9_{\pm0.5}$ | $33.2_{\pm0.6}$ | $38.9_{\pm0.7}$ | $\mathbf{49.4_{\pm0.1}}$ | ↑**10.5** |
| | $AP_{75}$ | – | – | – | – | – | – | $\mathbf{18.2_{\pm0.2}}$ | – |
| 2% | mAP | $12.4_{\pm0.4}$ | $13.8_{\pm0.3}$ | $10.9_{\pm0.6}$ | $10.4_{\pm0.4}$ | – | $23.9_{\pm0.5}$ | $\mathbf{26.3_{\pm0.1}}$ | ↑**2.4** |
| | $AP_{50}$ | $34.3_{\pm0.5}$ | $36.2_{\pm0.4}$ | $29.3_{\pm0.6}$ | $28.7_{\pm0.7}$ | – | $48.5_{\pm0.6}$ | $\mathbf{52.3_{\pm0.3}}$ | ↑**3.8** |
| | $AP_{75}$ | – | – | – | – | – | – | $\mathbf{23.5_{\pm0.1}}$ | – |

Table 3: **Results on COCO**. Our method $D^3$ consistently outperforms all baselines across different evaluation metrics. The last column indicates the performance gain over UniDD.

| Ratio | Metric | Random | Uniform | K-Center | Herding | DCOD | UniDD | $D^3$ (Ours) | Gain |
|---|---|---|---|---|---|---|---|---|---|
| 0.25% | mAP | $0.5_{\pm0.1}$ | $0.8_{\pm0.2}$ | $0.4_{\pm0.1}$ | $0.5_{\pm0.1}$ | – | $4.5_{\pm0.3}$ | $\mathbf{9.50_{\pm0.1}}$ | ↑**5.0** |
| | $AP_{50}$ | $1.7_{\pm0.3}$ | $2.4_{\pm0.5}$ | $1.5_{\pm0.2}$ | $1.8_{\pm0.4}$ | – | $10.3_{\pm0.4}$ | $\mathbf{19.9_{\pm0.3}}$ | ↑**9.6** |
| | $AP_{75}$ | – | – | – | – | – | – | $\mathbf{7.97_{\pm0.2}}$ | – |
| 0.5% | mAP | $3.7_{\pm0.2}$ | $3.4_{\pm0.4}$ | $3.2_{\pm0.5}$ | $3.5_{\pm0.3}$ | $3.7_{\pm0.3}$ | $7.1_{\pm0.4}$ | $\mathbf{11.9_{\pm0.1}}$ | ↑**4.8** |
| | $AP_{50}$ | $10.1_{\pm0.3}$ | $9.5_{\pm0.6}$ | $9.3_{\pm0.5}$ | $9.7_{\pm0.3}$ | $8.4_{\pm0.2}$ | $16.9_{\pm0.3}$ | $\mathbf{23.2_{\pm0.1}}$ | ↑**6.3** |
| | $AP_{75}$ | – | – | – | – | – | – | $\mathbf{10.8_{\pm0.1}}$ | – |
| 1% | mAP | $7.2_{\pm0.8}$ | $7.4_{\pm0.5}$ | $6.1_{\pm0.3}$ | $6.7_{\pm0.4}$ | $6.1_{\pm0.4}$ | $10.8_{\pm0.4}$ | $\mathbf{13.6_{\pm0.2}}$ | ↑**2.8** |
| | $AP_{50}$ | $17.3_{\pm0.9}$ | $17.6_{\pm0.5}$ | $15.4_{\pm0.6}$ | $16.3_{\pm0.7}$ | $13.5_{\pm0.3}$ | $22.5_{\pm0.5}$ | $\mathbf{25.2_{\pm0.3}}$ | ↑**2.7** |
| | $AP_{75}$ | – | – | – | – | – | – | $\mathbf{13.5_{\pm0.2}}$ | – |

## 4.4 SYNTHETIC DATA GENERATION PIPELINE

Our pipeline generates synthetic detection data through two main stages. First, we construct a scene-to-object dictionary $\mathcal{D}$ that captures dominant object co-occurrence patterns per scene category. Scene labels are predicted using a pretrained scene classifier, and object labels are aggregated across images sharing the same scene. Then, synthetic samples are generated as follows: (1) sample $m$ captions from the training set, (2) enrich each caption with scene-consistent object terms using $\mathcal{D}$, (3) merge them via LLM-based or region-aware composition into a single prompt, (4) generate an image from the prompt using a T2I model, and (5) apply a frozen object detector to obtain pseudo-labels by filtering high-confidence boxes. The resulting image–label pairs $(\tilde{x}, \mathcal{B})$ form the synthetic dataset used for training downstream detection models. We provide all prompts format in A.1.1.

## 5 EXPERIMENTS

### 5.1 EXPERIMENTAL SETUP

**Datasets and Metrics.** We evaluate $D^3$ on two widely-used object detection benchmarks: PASCAL VOC (Everingham et al., 2010) and MS COCO (Lin et al., 2014). All images are resized to $512 \times 512$ resolution during training and evaluation. There are 20 classes for VOC dataset and 80 classes for MS COCO dataset. For VOC, we follow the common practice(Qi et al., 2024) of merging the `trainval` sets from VOC 2007 and VOC 2012 (16,551 images) for training, and using VOC 2007 `test` for evaluation. For COCO, we adopt the standard setting of using 118,287 training images and 5,000 validation images for testing.

We report standard detection metrics: mean Average Precision (mAP), $AP_{50}$ (AP at IoU threshold 0.5), and $AP_{75}$ (AP at IoU threshold 0.75). For synthetic dataset evaluation, we assess the performance under limited training data. Specifically, we evaluate on 0.5%, 1%, and 2% of images for VOC, and 0.25%, 0.5%, and 1% of training data for COCO.

**Implementation Details.** If ground-truth captions are unavailable, we extract image-text pairs from the real dataset using BLIP(Li et al., 2022) to provide semantic descriptions of image content.

Table 4: **Performance on stronger detectors.**

| Ratio | Method | RTMDet | | | DINO | | | DiffusionDet | | |
|---|---|---|---|---|---|---|---|---|---|---|
| | | mAP | AP$_{50}$ | AP$_{75}$ | mAP | AP$_{50}$ | AP$_{75}$ | mAP | AP$_{50}$ | AP$_{75}$ |
| 0.25% | Random | $4.0_{\pm0.2}$ | $8.5_{\pm0.4}$ | $3.3_{\pm0.2}$ | $6.2_{\pm0.1}$ | $11.8_{\pm0.2}$ | $5.9_{\pm0.1}$ | $6.1_{\pm0.1}$ | $12.3_{\pm0.2}$ | $5.5_{\pm0.1}$ |
| | K-Center | $2.8_{\pm0.2}$ | $5.8_{\pm0.4}$ | $2.5_{\pm0.2}$ | $3.9_{\pm0.1}$ | $7.1_{\pm0.1}$ | $3.7_{\pm0.1}$ | $3.7_{\pm0.1}$ | $7.1_{\pm0.1}$ | $3.4_{\pm0.2}$ |
| | Herding | $3.7_{\pm0.4}$ | $7.8_{\pm0.8}$ | $3.2_{\pm0.4}$ | $6.1_{\pm0.2}$ | $11.2_{\pm0.3}$ | $5.7_{\pm0.3}$ | $5.8_{\pm0.1}$ | $11.6_{\pm0.1}$ | $5.2_{\pm0.2}$ |
| | D$^3$ (Ours) | $\mathbf{4.9}_{\pm0.3}$ | $\mathbf{9.7}_{\pm0.5}$ | $\mathbf{4.3}_{\pm0.4}$ | $\mathbf{9.2}_{\pm0.5}$ | $\mathbf{17.3}_{\pm0.8}$ | $\mathbf{8.8}_{\pm0.7}$ | $\mathbf{11.3}_{\pm0.1}$ | $\mathbf{20.6}_{\pm0.1}$ | $\mathbf{11.0}_{\pm0.1}$ |
| 0.5% | Random | $8.3_{\pm0.3}$ | $16.2_{\pm0.5}$ | $7.6_{\pm0.3}$ | $9.6_{\pm0.2}$ | $17.3_{\pm0.4}$ | $9.2_{\pm0.3}$ | $9.3_{\pm0.1}$ | $17.4_{\pm0.2}$ | $8.8_{\pm0.2}$ |
| | K-Center | $5.8_{\pm0.2}$ | $11.1_{\pm0.4}$ | $5.3_{\pm0.2}$ | $6.2_{\pm0.1}$ | $10.8_{\pm0.2}$ | $5.9_{\pm0.1}$ | $5.9_{\pm0.1}$ | $10.9_{\pm0.1}$ | $5.6_{\pm0.1}$ |
| | Herding | $8.3_{\pm0.1}$ | $15.8_{\pm0.1}$ | $7.6_{\pm0.2}$ | $10.0_{\pm0.1}$ | $17.8_{\pm0.2}$ | $9.7_{\pm0.2}$ | $9.2_{\pm0.1}$ | $17.4_{\pm0.1}$ | $8.7_{\pm0.1}$ |
| | D$^3$ (Ours) | $\mathbf{10.2}_{\pm0.1}$ | $\mathbf{18.7}_{\pm0.3}$ | $\mathbf{9.9}_{\pm0.2}$ | $\mathbf{12.7}_{\pm0.2}$ | $\mathbf{22.7}_{\pm0.2}$ | $\mathbf{13.0}_{\pm0.2}$ | $\mathbf{14.0}_{\pm0.4}$ | $\mathbf{24.2}_{\pm0.5}$ | $\mathbf{14.2}_{\pm0.4}$ |
| 1% | Random | $13.2_{\pm0.2}$ | $23.6_{\pm0.4}$ | $13.1_{\pm0.3}$ | $13.4_{\pm0.2}$ | $22.7_{\pm0.1}$ | $13.5_{\pm0.2}$ | $12.9_{\pm0.1}$ | $22.6_{\pm0.1}$ | $12.6_{\pm0.1}$ |
| | K-Center | $9.4_{\pm0.7}$ | $17.0_{\pm0.9}$ | $9.1_{\pm1.0}$ | $9.2_{\pm0.1}$ | $15.6_{\pm0.1}$ | $9.1_{\pm0.1}$ | $8.8_{\pm0.2}$ | $15.4_{\pm0.2}$ | $8.7_{\pm0.3}$ |
| | Herding | $13.3_{\pm0.2}$ | $23.6_{\pm0.3}$ | $13.2_{\pm0.3}$ | $13.2_{\pm0.1}$ | $22.7_{\pm0.2}$ | $13.2_{\pm0.1}$ | $12.9_{\pm0.1}$ | $22.8_{\pm0.1}$ | $12.8_{\pm0.1}$ |
| | D$^3$ (Ours) | $\mathbf{14.7}_{\pm0.2}$ | $\mathbf{25.5}_{\pm0.5}$ | $\mathbf{15.0}_{\pm0.2}$ | $\mathbf{15.9}_{\pm0.2}$ | $\mathbf{27.2}_{\pm0.1}$ | $\mathbf{16.7}_{\pm0.2}$ | $\mathbf{16.4}_{\pm0.1}$ | $\mathbf{27.7}_{\pm0.1}$ | $\mathbf{16.9}_{\pm0.2}$ |

To construct scene-to-object dictionary, we use a ResNet-50 pretrained on Places365 (Zhou et al., 2017) to classify the scene and build the dictionary. For caption enrichment, we employ GPT-4.1 (OpenAI, 2024) as a language model to inject additional, scene-relevant object mentions while maintaining semantic coherence. For multi-caption composition, we adopt a region-aware strategy, which offers a simple yet effective means of increasing object diversity and spatial coverage in generated images. We adopt Stable Diffusion v3.0 (SDv3) (Stability AI, 2024) as our T2I model to synthesize high-resolution images. To generate pseudo ground-truth annotations, we apply a pretrained Faster R-CNN detector(Ren et al., 2015) to the synthesized image and retain detections with confidence scores above a threshold of $\tau = 0.3$. For evaluation, we train Faster R-CNN 50 from scratch using three random seeds and report the mean and standard deviation of performance metrics across runs. All experiments are conducted on a single NVIDIA RTX 3090 GPU.

## 5.2 MAIN RESULTS

**Comparison to SOTA Methods.** We compare our method with core-set selection baselines, including Random(Chen et al., 2010), K-Center(Tsang et al., 2005), and Herding(Castro et al., 2018), as well as recent condensation methods DCOD(Qi et al., 2024) and UniDD(Qi et al., 2025). On VOC, as shown in Table 2, our approach achieves strong performance, especially under extreme data constraints. Using only 0.5% of data, we reach 18.8% mAP, outperforming UniDD by 10.3 points (8.5% mAP). At a 1% data budget, our method maintains a 6.1-point gain, and still surpasses UniDD at 2% of data, demonstrating consistent improvement across data regimes.

On the more challenging COCO benchmark, our method consistently outperforms UniDD across all tested budgets. Notably, we obtain a 5.0-point gain at a 0.25% of data and 4.8-point gain at 0.5% of data, as shown in Table 3. These results highlight the effectiveness of our framework in generating compact yet high-quality synthetic datasets for object detection.

**Results on Other Detectors.** To assess the generality of our synthetic dataset, we evaluate its effectiveness across multiple object detection architectures. This extends beyond prior works such as DCOD and UniDD, which primarily report results only on Faster R-CNN. Specifically, we adopt RTMDet(Lyu et al., 2022), DINO(Zhang et al., 2023), and DiffusionDet(Chen et al., 2023) as strong backbones and conduct experiments on the COCO dataset. Since publicly available code for DCOD and UniDD is not accessible, we instead compare against coreset selection baselines under the same training setup. As shown in Table 4, our method consistently outperforms coreset baselines across all data budgets. For instance, at 0.25% of COCO data, random selection achieves 6.2% mAP with DINO and 6.1% mAP with DiffusionDet, while our method achieves 9.2% and 11.3% mAP, respectively. At 1% data, DiffusionDet achieves the best performance with 16.4% mAP, followed by DINO (15.9% mAP) and Faster R-CNN (13.6% mAP), demonstrating the general effectiveness of our synthetic data across diverse detector types.

**Generalization to Other Tasks.** We further assess our framework's versatility beyond detection by evaluating it on semantic segmentation and pose estimation. For segmentation we use MobileNet-V3 as backbone and LR-ASPP as segmentation head (Howard et al., 2019); for pose estimation task, we use a ResNet-50 backbone (Yu et al., 2021). As summarized in Table 5, $D^3$ consistently outperforms Random, K-Center, Herding, and UniDD across all budgets: 15.8 mIoU at 3.5% (+5.5 over UniDD's 10.3), 17.9 at 7% (+5.6), and 20.0 at 14% (+5.1).

On AP-10K dataset (Yu et al., 2021) (Table 6), $D^3$ reaches 21.7 mAP at 1% (+14.6 over the best selector), and maintains clear gains at 2% (27.9 vs. 18.4, +9.5) and 4% (33.4 vs. 28.6, +4.8). These improvements—obtained with lightweight students—indicate that our scene-aware enrichment primarily boosts data quality rather than relying on model capacity, and is especially effective under tight budgets.

Table 5: **Results on segmentation task using VOC dataset.**

| Method | Ratio: 3.5% | Ratio: 7% | Ratio: 14% |
|---|---|---|---|
| Random | $6.7_{\pm 0.4}$ | $9.6_{\pm 0.5}$ | $11.3_{\pm 0.4}$ |
| K-Center | $6.1_{\pm 0.4}$ | $8.5_{\pm 0.6}$ | $10.4_{\pm 0.7}$ |
| Herding | $6.2_{\pm 0.7}$ | $8.7_{\pm 0.4}$ | $10.6_{\pm 0.3}$ |
| UniDD | $10.3_{\pm 0.5}$ | $12.3_{\pm 0.3}$ | $14.9_{\pm 0.7}$ |
| $D^3$ (Ours) | $\mathbf{15.8_{\pm 0.2}}$ | $\mathbf{17.9_{\pm 0.1}}$ | $\mathbf{20.0_{\pm 0.8}}$ |

Table 6: **Results on animal pose estimation task.**

| Method | Ratio: 1% | Ratio: 2% | Ratio: 4% |
|---|---|---|---|
| Random | $7.1_{\pm 0.6}$ | $15.9_{\pm 0.4}$ | $28.6_{\pm 0.1}$ |
| K-Center | $4.8_{\pm 0.2}$ | $13.8_{\pm 0.5}$ | $25.2_{\pm 0.5}$ |
| Herding | $6.2_{\pm 0.1}$ | $18.4_{\pm 0.4}$ | $28.0_{\pm 0.1}$ |
| $D^3$ (Ours) | $\mathbf{21.7_{\pm 0.36}}$ | $\mathbf{27.9_{\pm 0.6}}$ | $\mathbf{33.4_{\pm 0.6}}$ |

## 5.3 ABLATION STUDY

We conduct a comprehensive ablation study to assess the impact of each component in our pipeline, including module contributions, pseudo-labeling detectors, cross-dataset caption transferability, and visualizations of synthetic data.

Table 7: **Ablation study of the proposed components.** $MCC_{LLM}$: LLM multi-caption composition. $MCC_{RA}$: region-aware composition.

| Baseline | SACE | $MCC_{LLM}$ | $MCC_{RA}$ | mAP |
|---|---|---|---|---|
| ✓ | ✗ | ✗ | ✗ | $4.5_{\pm 0.1}$ |
| ✓ | ✓ | ✗ | ✗ | $6.8_{\pm 0.1}$ |
| ✓ | ✗ | ✓ | ✗ | $6.9_{\pm 0.1}$ |
| ✓ | ✗ | ✗ | ✓ | $8.3_{\pm 0.1}$ |
| ✓ | ✓ | ✗ | ✓ | $\mathbf{9.5_{\pm 0.1}}$ |

Table 8: **Pseudo-labeler vs. evaluator on COCO (0.25% of data).** Synthetic images are labeled by model **L** and evaluated with model **E**.

| E\L | F R-CNN | DINO | DiffusionDet |
|---|---|---|---|
| F R-CNN | $9.5_{\pm 0.1}$ | $8.3_{\pm 0.1}$ | $8.5_{\pm 0.2}$ |
| DINO | $9.2_{\pm 0.5}$ | $9.1_{\pm 0.2}$ | $8.8_{\pm 0.2}$ |
| DiffusionDet | $11.3_{\pm 0.1}$ | $10.0_{\pm 0.1}$ | $9.6_{\pm 0.2}$ |

**Effectiveness of Each Component.** We report the performance of each component on COCO at 0.25% of data in Table 7. Starting with SDv3 alone (baseline), our method achieves 4.5% mAP. Applying SACE improves the performance to 6.8%. Introducing multi-caption composition using a language model improves the performance to 6.9% mAP. When replacing LLM-based composition with the region-aware strategy, we observe a substantial boost to 8.3% mAP. Finally, incorporating the SACE module to region-aware composition reach the highest performance of 9.5% mAP, which is more than doubling the initial result.

**Impact of Pseudo-Labeling Quality.** Table 8 compares detectors used to generate pseudo-labels for synthetic COCO images. Across evaluators, pseudo-labels from Faster R-CNN consistently yield the best downstream mAP (e.g., DiffusionDet reaches 11.3% with Faster R-CNN labels vs. 10.0% with DINO and 9.6% with DiffusionDet), indicating that a stronger model does not necessarily translate into better supervision in this setting. Thus, improving the quality of the label is a promising research direction.

**Cross-Dataset Generalization.** We investigate the effectiveness of using captions from a source dataset to generate synthetic images for a different target dataset, facilitating cross-domain transfer of textual supervision. As shown in Table 9, when using COCO captions to generate images evaluated on VOC, our method consistently outperforms the baseline and remains competitive with in-domain generation, particularly as the training budget increases.

Table 9: **Cross-dataset generalization results**. Baseline indicates the original caption.

| Source → Target | Method | 0.5% | 1% | 2% |
|---|---|---|---|---|
| VOC → VOC | $D^3$ | $18.8_{\pm 0.1}$ | $22.9_{\pm 0.1}$ | $26.3_{\pm 0.1}$ |
| COCO → VOC | Baseline | $8.2_{\pm 0.4}$ | $13.3_{\pm 0.6}$ | $20.8_{\pm 0.2}$ |
| | $D^3$ | $\mathbf{14.0_{\pm 0.3}}$ | $\mathbf{19.7_{\pm 0.2}}$ | $\mathbf{23.9_{\pm 0.3}}$ |

Table 10: **SACE is robust to LLM choice.**

| Model | mAP |
|---|---|
| GPT-4.1 (default) | $9.5_{\pm 0.1}$ |
| Gemini-1.5-Pro | $9.3_{\pm 0.2}$ |
| Claude-3.7-Sonnet | $9.6_{\pm 0.2}$ |
| Qwen2.5-7B | $9.4_{\pm 0.2}$ |

Table 11: **Caption extractor ablation.**

| Method | mAP |
|---|---|
| BLIP | $18.8_{\pm 0.1}$ |
| LLAVA | $18.4_{\pm 0.1}$ |

| Baseline | SACE | $D^3$ |
|---|---|---|
| Caption: A **boy** riding a **skateboard** is doing tricks. | Caption: A **boy** wearing a **backpack** rides a **skateboard** doing tricks near a parked **car** and a **stop sign**, with a **potted plant** and a **chair** on the sidewalk. | Caption: Top Left: A bedroom with a canopy **bed**, dresser, and lamp, where a **person** sits on the **bed** reading a **book** beside a **potted plant** and a **cup** on a **chair**. Top Right: A cutting board with stuffed birds sits on a **dining table**, beside a sharp **knife** and a nearby… |
| Synthetic image / Synthetic image + bounding boxes | Synthetic image / Synthetic image + bounding boxes | Synthetic image / Synthetic image + bounding boxes |

Figure 5: **Baseline vs. enriched captions.** The baseline caption provides minimal context and few object categories, whereas our methods enrich scene semantics and increase per-image object diversity. Objects referenced in the captions are highlighted in **bold black**.

**Choice of LLM for SACE.** As shown in Table 10, $D^3$ is largely LLM-agnostic for SACE: all models land within 9.3–9.6 mAP, with differences on the order of the reported variance (±0.1–0.2). This indicates our enrichment pipeline—not a particular LLM—drives the gains; we use GPT-4.1 by default, but comparable performance is obtainable with alternative LLMs.

**Choice of Caption Extractor.** We compare BLIP (Li et al., 2022) and LLaVA (Liu et al., 2023) under identical settings (same prompt) on VOC at the 0.5% budget. As shown in Table 11, BLIP attains 18.8 mAP vs. 18.4 mAP for LLaVA. This suggests that our pipeline is largely robust to the caption extractor. We default to BLIP for convenience.

**Visualization and Object Diversity.** On COCO (Figure 5), SACE (col. 2) adds context-appropriate objects missing from the original caption (e.g., car, stop sign, backpack, chair), while our region-aware multi-caption composition (last col.) assigns sub-captions to specific areas (e.g., top-left/right), promoting diverse, spatially distributed content. To complement these visual comparisons, we report the average number of unique object classes per image under a 0.5% storage budget for COCO and 1% for VOC. As shown in Table 12, our method significantly outperforms the random selection, achieving approximately $5.9\times$ increase in COCO dataset, which demonstrates the effectiveness of our design in promoting semantic richness and object diversity in the generated images.

Table 12: **Unique categories per image.**

| Method | VOC | COCO |
|---|---|---|
| Random | 1.5 | 2.3 |
| $MCC_{LLM}$ | 4.0 | 6.2 |
| $MCC_{RA}$ | 4.3 | 6.9 |
| $D^3$ | **7.7** | **13.5** |

## 6 CONCLUSION

We introduce $D^3$, the first framework to leverage scene-aware semantics and structured prompt composition for dataset condensation in object detection. By combining a Scene-to-Object Dictionary and multi-caption composition, our method synthesizes semantically rich, multi-object images using frozen T2I models. Extensive experiments show that $D^3$ synthesizes images with semantically dense and significantly outperforms prior condensation and core-set selection methods under extremely constrained data budgets, with strong generalization across detectors and datasets.

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

## A  APPENDIX

### A.1  EXPERIMENTAL DETAILS

#### A.1.1  PROMPTS USED IN OUR FRAMEWORK

To extract captions, we adopt pre-trained vision–language models (VLMs) such as BLIP (Li et al., 2022) and LLaVA (Liu et al., 2023), using with a single instruction to ensure a fair, comparable setup.

---

**Captioning Prompt**

**Instruction:** Describe the image in detail.

---

We provide the prompts used to guide the language model (i.e., GPT-4.1OpenAI (2024)) in our framework, specifically for Scene-Aware Caption Enrichment and MCC-LLM Prompt Merging.

---

**Prompt for Scene-Aware Caption Enrichment**

**Original Caption:** {caption}
**Object List:** {possible_objects}
**Instruction:** Please enrich the caption by smoothly incorporating as many contextually appropriate objects from the list as possible. The result should be a single fluent sentence under 30 words, suitable for Stable Diffusion image generation.

---

**Prompt for MCC-LLM Merging**

**Original Captions:** {A list of captions}
**Instruction:** Merge the above captions into a single sentence that sounds natural and fluent. Keep it under 100 words and make sure the result reads like a single scene or connected description.

---

### A.1.2 ARCHITECTURES USED FOR EVALUATION

We evaluate our method using several representative architectures:

- **Faster R-CNN**(Ren et al., 2015): A two-stage detector that first generates region proposals and then classifies and refines them.
- **DINO**(Zhang et al., 2023): A transformer-based detector built on DETR (Carion et al., 2020), enhancing object query learning and convergence speed.
- **DiffusionDet**(Chen et al., 2023): A detector that formulates object detection as a denoising diffusion process, offering strong performance without hand-designed priors.
- **RTMDet**(Lyu et al., 2022): A recent one-stage detector optimized for real-time and high-accuracy detection, using advanced architectural designs and training strategies.
- **MobileNet-V3** (Howard et al., 2019): An efficient network targerted for low latency. We adopt this network as the feature extractor and use LR-ASPP as the segmentation head. This network is used to evaluate on segmentation task.
- **ResNet-50** (He et al., 2015): A deep neural network with residual connection. This network is used for the animal pose estimation task.

## A.2 ADDITIONAL EXPERIMENTS

### A.2.1 COPY-PASTE WITH MORE TRAINING BUDGET

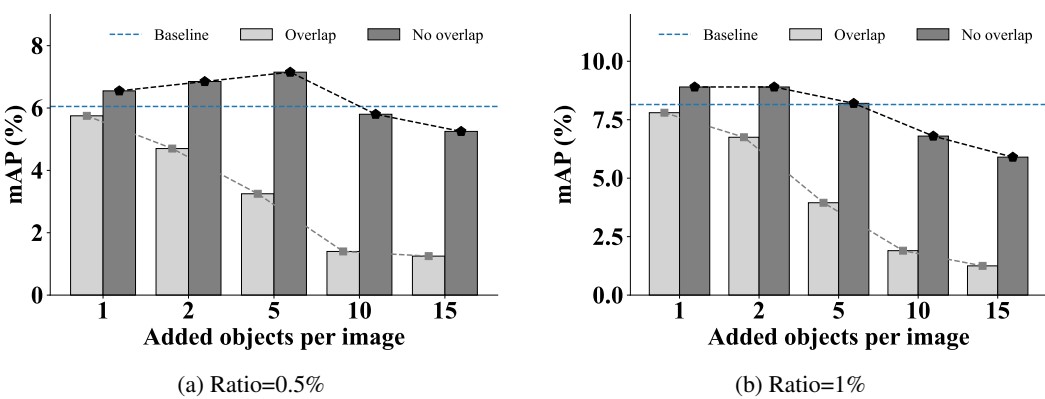

(a) Ratio=0.5%           (b) Ratio=1%

Figure 6: **Results on COCO dataset**.

We conduct additional experiments using Copy-Paste (Ghiasi et al., 2021) to augment the training data. As shown in Figure 6, we observe a consistent pattern: with non-overlapping placement, adding more pasted objects yields initial gains, after which the improvement saturates and can eventually decline. In contrast, when overlaps are allowed, increasing the number of pasted objects monotonically degrades performance.

### A.2.2 COMPARISON TO STRONGER CAPTIONING MODELS

Table 13: **Ablation on difference captioning models.**

| Dataset | ControlCap | Qwen3-VL | BLIP | BLIP+SACE |
|---------|-----------|----------|------|-----------|
| VOC | $8.3_{\pm0.1}$ | $8.2_{\pm0.4}$ | $8.2_{\pm0.1}$ | $\mathbf{14.5_{\pm0.2}}$ |
| COCO | $4.1_{\pm0.1}$ | $4.5_{\pm0.1}$ | $4.9_{\pm0.2}$ | $\mathbf{6.8_{\pm0.1}}$ |

We perform additional experiments using two stronger captioners, namely ControlCap (Zhao et al., 2025b) and Qwen3-VL (Bai et al., 2025). In all cases, the resulting captions were used to synthesize images with the same SD backbone. We further applied our SACE module on top of BLIP captioner.

As shown in the Table 13, using stronger captioners alone produces performance comparable to BLIP (only a +0.1–0.4 AP difference). In contrast, enriching the BLIP captions with SACE yields a substantially larger improvement (from 8.2 → 14.3 AP on VOC). This indicates that SACE provides complementary benefits beyond what modern captioners offer, and that the gains are not solely due to captioner strength.

### A.2.3 ABLATION ON NUMBER OF REGIONS

We study the effectiveness of our framework by varying the number of regions. Specifically, we evaluate layouts of $1\times2$ (vertical split), $2\times1$ (horizontal split), $2\times2$ (4 regions), $3\times3$ (9 regions), and $4\times4$ (16 regions).

Table 14: **Ablation on number of regions**. The experiments are performed on the VOC at 0.5% of data.

| Number of regions | 2 ($1\times2$) | 2 ($2\times1$) | 4 ($2\times2$) | 9 ($3\times3$) | 16 ($4\times4$) |
|---|---|---|---|---|---|
| mAP(%) | $13.4_{\pm0.2}$ | $15.2_{\pm0.3}$ | $\mathbf{18.8_{\pm0.1}}$ | $9.8_{\pm0.4}$ | $5.3_{\pm0.4}$ |

As shown in Table 14, using more regions decreases performance. There are two possible reasons for this. First, caption-length constraints: including 9–16 sub-captions exceeds the maximum token capacity of the text encoder, causing truncation and weakening the conditioning signal. Second, resolution constraints: with nine or more regions, each sub-image becomes approximately $170\times170$ pixels or smaller, which is insufficient for SD to generate fine object details. Due to these reasons, we default to using 4 regions as the most practical and stable configuration.

### A.2.4 IMPACT OF TEXT-TO-IMAGE BACKBONE (SDv1.5 vs SDv3)

Table 15: **Performance of our framework using SDv1.5.**

| Method | mAP (%) |
|---|---|
| UniDD | $8.5_{\pm0.4}$ |
| $D^3$ with SDv1.5 | $11.4_{\pm0.2}$ |
| $D^3$ with SDv3 | $18.8_{\pm0.1}$ |

We report the performance of our framework using SDv1.5 on VOC dataset at 0.5% of data in Table 15. Our framework achieves an improvement of approximately 3% mAP over UniDD, indicating that the proposed components provide benefits beyond the choice of backbone. Since our input prompts are more complex than the base captions, a stronger text encoder naturally helps produce more informative conditioning features. This explains why using SDv3 yields larger gains.

### A.2.5 IMPACT OF SAMPLING STEPS IN STABLE DIFFUSION

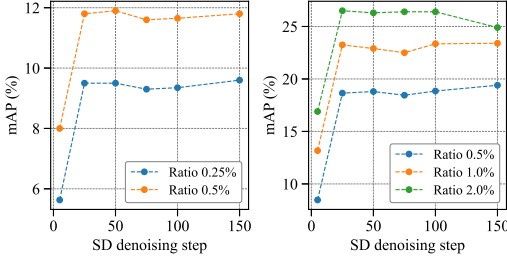

Figure 7: **Sensitivity to sampling steps.** mAP (%) vs. number of steps on COCO (left) and VOC (right); higher is better.

We study the effect of the number of denoising steps used in SD during image synthesis, as shown in Figure 7. With very few steps (e.g., 10–15), the generated images lack sufficient quality, resulting in lower detection performance. Performance steadily improves as the number of steps increases, and stabilizes beyond 25 steps. By default, we use 50 denoising steps across all experiments.

### A.2.6 DISTILLATION COST

Table 16: **Computation time for inference** (in seconds).

| Category | Run 1 | Run 2 | Run 3 | Average |
|---|---|---|---|---|
| Scene prediction | 0.0845 | 0.0843 | 0.0854 | 0.08473 |
| SACE (GPT call) | 2.04 | 0.88 | 1.09 | 1.34 |
| Generation (T2I call) | 27.40 | 33.20 | 33.16 | 31.25 |

**SOD construction cost.** Constructing SOD requires a pretrained scene classifier. We use a ResNet-50 pretrained on Places365 dataset to compute SOD. A single forward pass takes approximately 0.084 s per image; at this rate, generating SOD for VOC ($\sim$16.5k images) takes about 23 min, and for COCO ($\sim$118k images) about 2.77 h.

**Inference Cost.** We report the inference cost of each module in Table 16, where generating a single image takes approximately 32.67 seconds. Although UniDD Qi et al. (2025) does not explicitly provide inference costs for object detection, it requires 250 sampling steps, which is substantially higher than our 50 steps, and therefore expected to incur a longer runtime.

In terms of memory usage, our approach based on SDv3 consumes only 5,296 MB of GPU memory, which is significantly lower than the 12.5 GB required by UniDD.

### A.3 SCENE-AWARE CAPTION ENRICHMENT EXAMPLES

To demonstrate the effectiveness of our scene-aware caption enrichment strategy, we present representative examples from the COCO and VOC datasets. We highlight objects that appear in the enriched captions using **bolded black** for clarity.

### A.3.1 ON MS COCO DATASET:

---

**Example 1**

*Original:* A spoon, a sandwich with some lettuce, and a drink.
*Enriched:* A **sandwich** with lettuce sits beside a **spoon** and a drink, while a curious **dog** eyes a **donut** on the table near a **cell phone**.

---

**Example 2**

*Original:* The people are sitting in a restaurant with planes hanging from the ceiling.
*Enriched:* Several **people** are sitting in a restaurant with planes hanging from the ceiling, some checking their **cell phones** while a **handbag** and a **suitcase** rest beside their table.

---

**Example 3**

*Original:* A group riding horses along a trail on the beach.
*Enriched:* A group rides **horses** along a sandy beach trail, passing a weathered **parking meter** near a bright red **fire hydrant** with a distant **clock** tower visible.

---

**Example 4**

*Original:* A soccer field with a group of people lined up on the field.
*Enriched:* On a soccer field, a group of **people** lined up near a **bench** and some **chairs**, with a **sports ball** and a few **bottles** scattered nearby.

---

### A.3.2   ON PASCAL VOC DATASET:

> **Example 1**
> *Original:* Two dogs are eating grass in a grassy area.
> *Enriched:* Two **dogs** are eating grass near a **cow** while a **bird** perches on a **bicycle** beside a **potted plant** in the grassy area.

> **Example 2**
> *Original:* A man and a woman cutting a wedding cake.
> *Enriched:* A **man and a woman** cut their wedding cake at a **dining table** decorated with **potted plants**, while a **cat** sits nearby on a **chair**.

> **Example 3**
> *Original:* Two children are sitting on a wooden chair.
> *Enriched:* Two **children** are sitting on a wooden **chair** next to a lazy **dog** while a **cat** lounges nearby and a **bicycle** rests against the wall.

> **Example 4**
> *Original:* A table with a lot of food on it.
> *Enriched:* A table covered with lots of food sits in front of a **sofa** and **chairs**, with a **person** holding a **bottle** while a **dog** lounges nearby.

### A.4   VISUALIZATION

To qualitatively evaluate the effectiveness of our Scene-Aware Caption Enrichment strategy, we visualize samples from the VOC and COCO datasets paired with both the original and enriched captions. As shown in Figure 8 and Figure 9, SACE consistently introduces scene-relevant objects from the VOC/COCO label space while maintaining fluency and coherence. These enriched captions offer more descriptive and contextually aligned textual guidance, which supports the generation of semantically richer images.

Additionally, we present representative samples synthesized by SDv3 using (i) the baseline captions (Figure 10), (ii) prompt merging via LLMs (Figure 11), and (iii) our full $D^3$ framework (Figure 12) on COCO. For VOC, we compare the baseline (Figure 13) and $D^3$-generated images (Figure 14) to highlight improvements in visual and semantic fidelity.

### A.5   DISCUSSION, LIMITATIONS, AND FUTURE WORK

While our framework provides better performance compared to previous works, there are several potential directions for further exploration. First, as shown in Table 8, pseudo-labeling networks affect the performance. This suggests that developing a better algorithm to label the synthetic data is crucial for improving the quality of synthetic images and their labels. Second, naively selecting top-$k$ objects to enrich the base caption may amplify the class imbalance. Thus, handling class balance or boosting rare classes is an interesting research direction. Third, explicitly controlling the object placement, location, and scale via GLIGEN(Li et al., 2023) or ControlNet(Zhang et al.) may be better than purely relying on SD prior. This can be seen as an extension for better localization of the generation process. Fourth, improving the quality of dataset condensation and developing dataset condensation methods that work for mid- and low-compression ratios is also an important research direction.

**Limitations.** Our approach is constrained by Stable Diffusion's prompt-length limit. Because region-aware composition is specified textually, we currently restrict the design to four regions. Extending to finer partitions (e.g., $3 \times 3$ or $4 \times 4$; 9 or 16 regions) would require longer prompts that SD truncates. Future work could leverage text-to-image models with larger context windows or structured conditioning (e.g., layout/segmentation guidance) to support more regions without truncation. Another limitation of our method is that it requires a strong text encoder such as CLIP L/14 or T5-XXL to parse complex prompts. Using a weak text encoder may hinder the ability of SD to generate proper images given complex prompts.

## A.6 LLM Usage

An AI assistant (ChatGPT) was employed exclusively for minor language refinement. The research conception, methodology, experiment, and analysis were conducted by the authors.

| | Text prompt given to SDv3 | Synthetic images | Synthetic images + bounding boxes |
|---|---|---|---|
| Baseline | An airplane is flying through a cloudy sky. | | |
| SACE | An airplane soars through a cloudy sky as birds fly nearby, while below, a train and a few cars travel past a parked bus. | | |
| Baseline | A girl is painting a picture on a easel in a living room. | | |
| SACE | A girl paints at an easel in a cozy living room, with a cat lounging on the sofa and a tvmonitor glowing nearby. | | |

Figure 8: **SACE on VOC:** enriched captions add scene-relevant objects and improve contextual alignment.

| Text prompt given to SDv3 | Synthetic images | Synthetic images + bounding boxes |
|---|---|---|
| *Baseline* — A plate with churros and hot chocolate on a wooden table. | | |
| *SACE* — A plate with churros and hot chocolate sits on a wooden table beside a wine glass, a book, and a bowl with a carrot. | | |
| *Baseline* — Two men in suits and ties next to plant. | | |
| *SACE* — Two men in suits and ties stand next to a plant, with a suitcase at their feet, a laptop open on a table, and a wine glass nearby. | | |

Figure 9: **SACE on COCO:**. The enriched captions generated by SACE successfully incorporate scene-relevant COCO objects, enhancing contextual richness and alignment with the visual content.

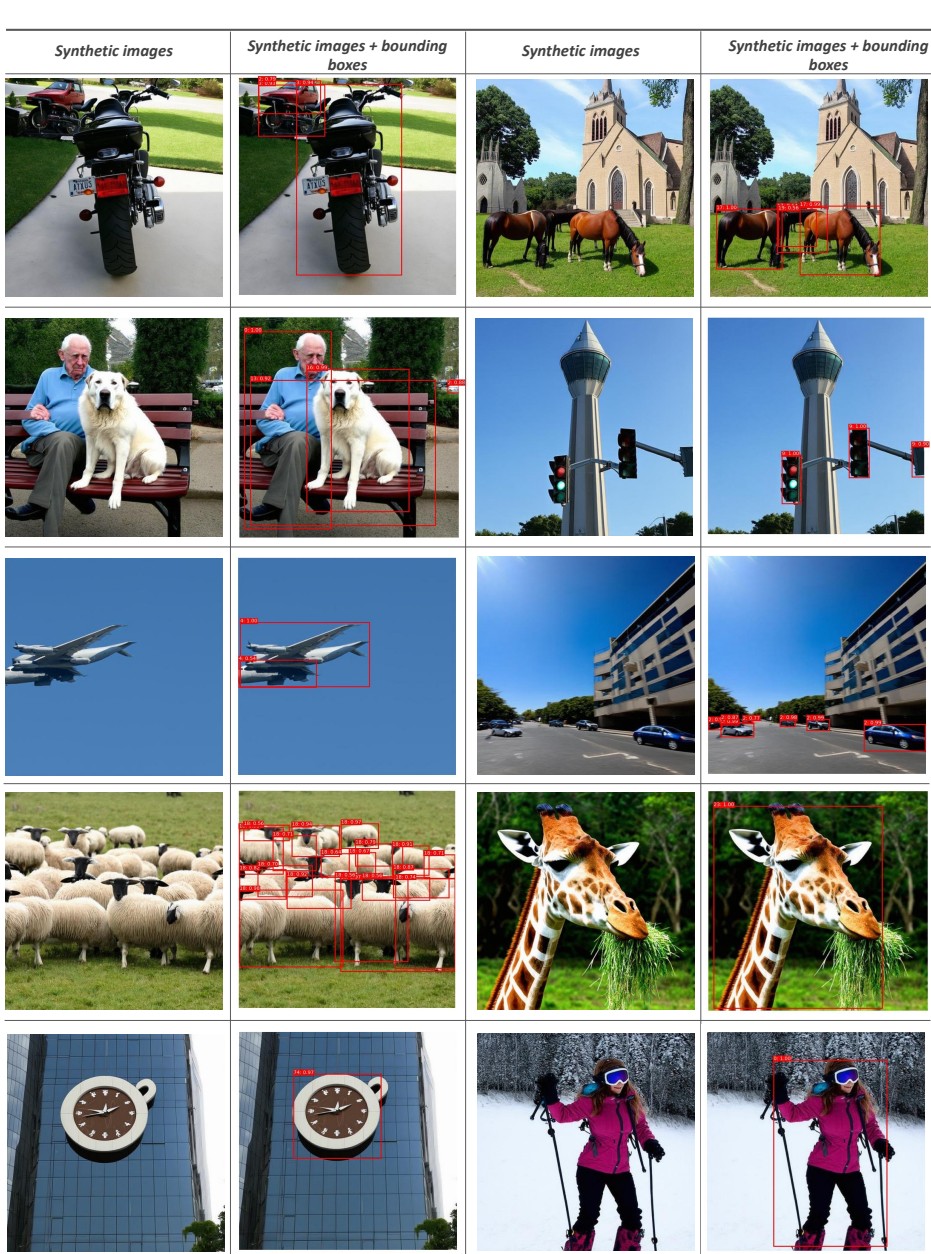

Figure 10: **Generation from baseline captions (COCO).** Representative samples.

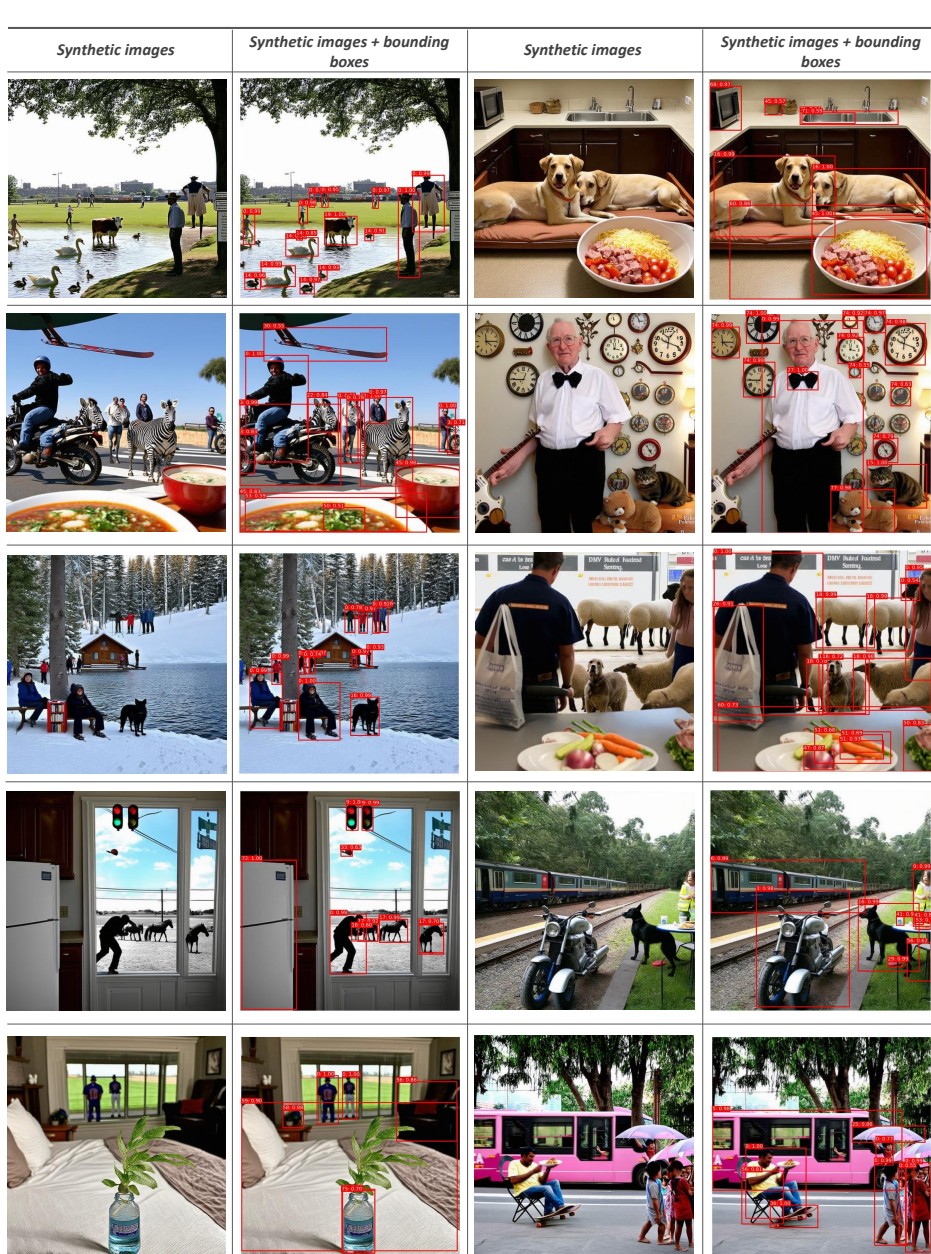

Figure 11: **MCC–LLM caption merging on COCO.** Selected synthesized images.

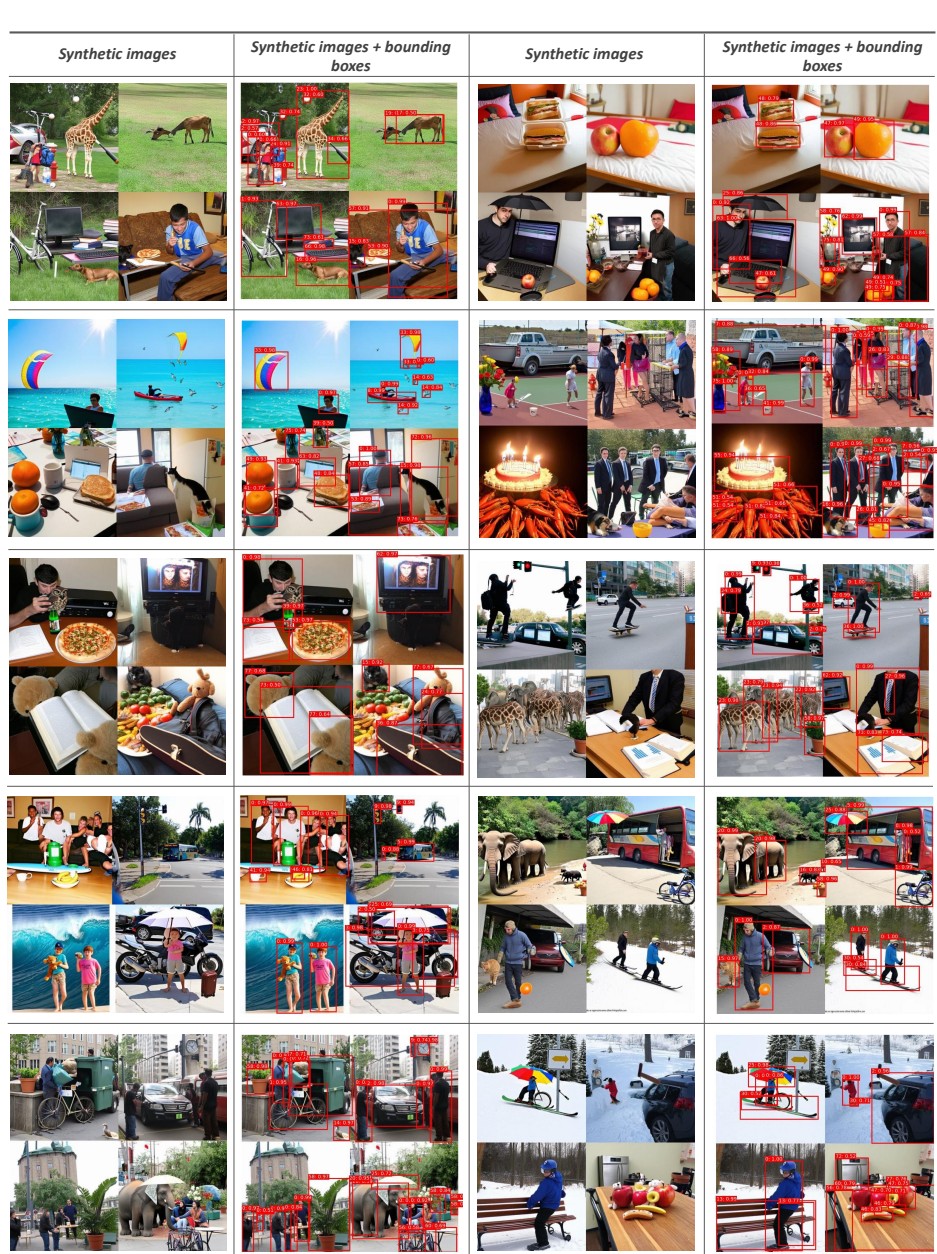

Figure 12: **Generated images on COCO using** $D^3$**.** Representative samples.

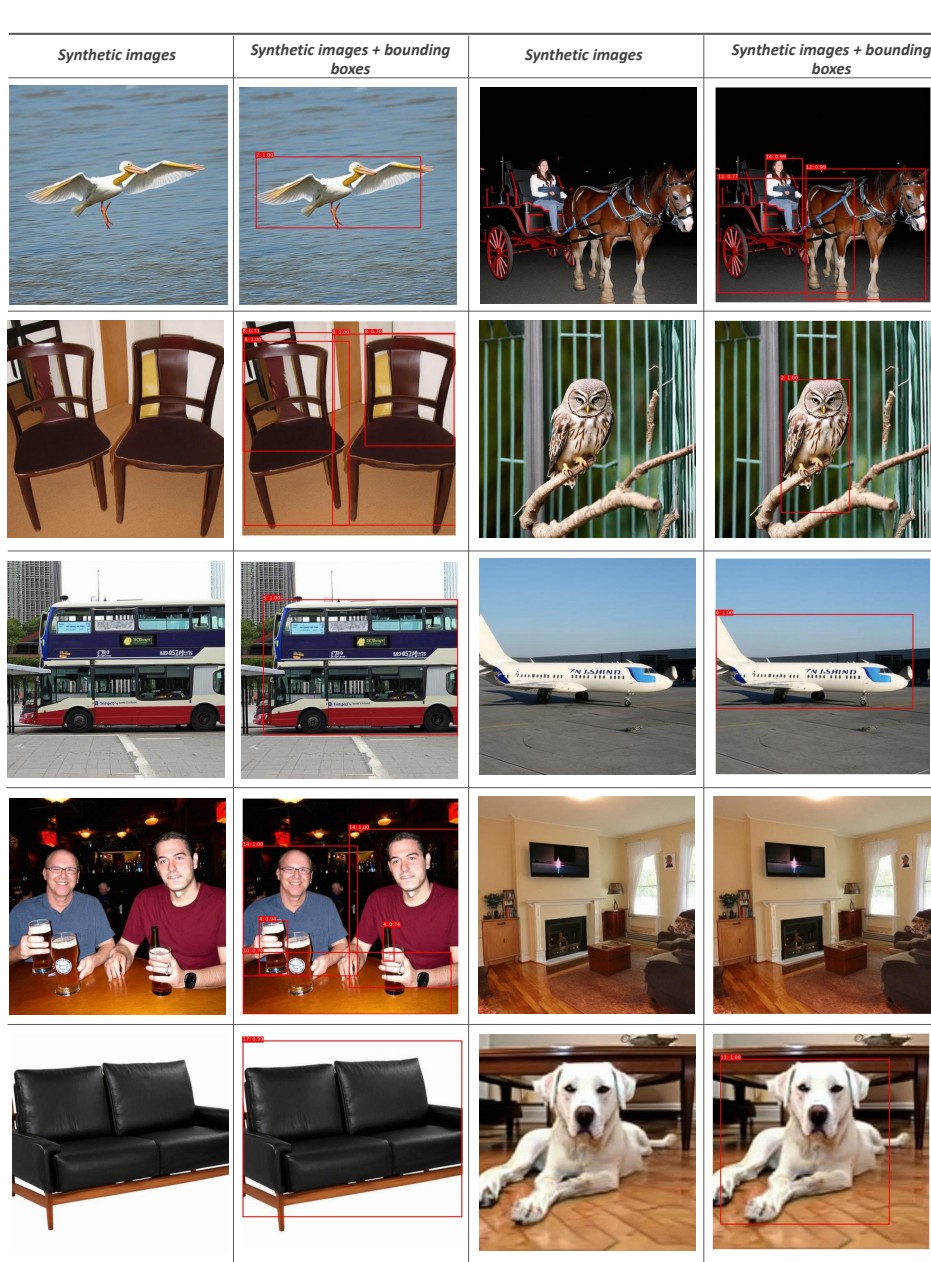

Figure 13: **Generation from baseline captions (VOC).** Representative samples.

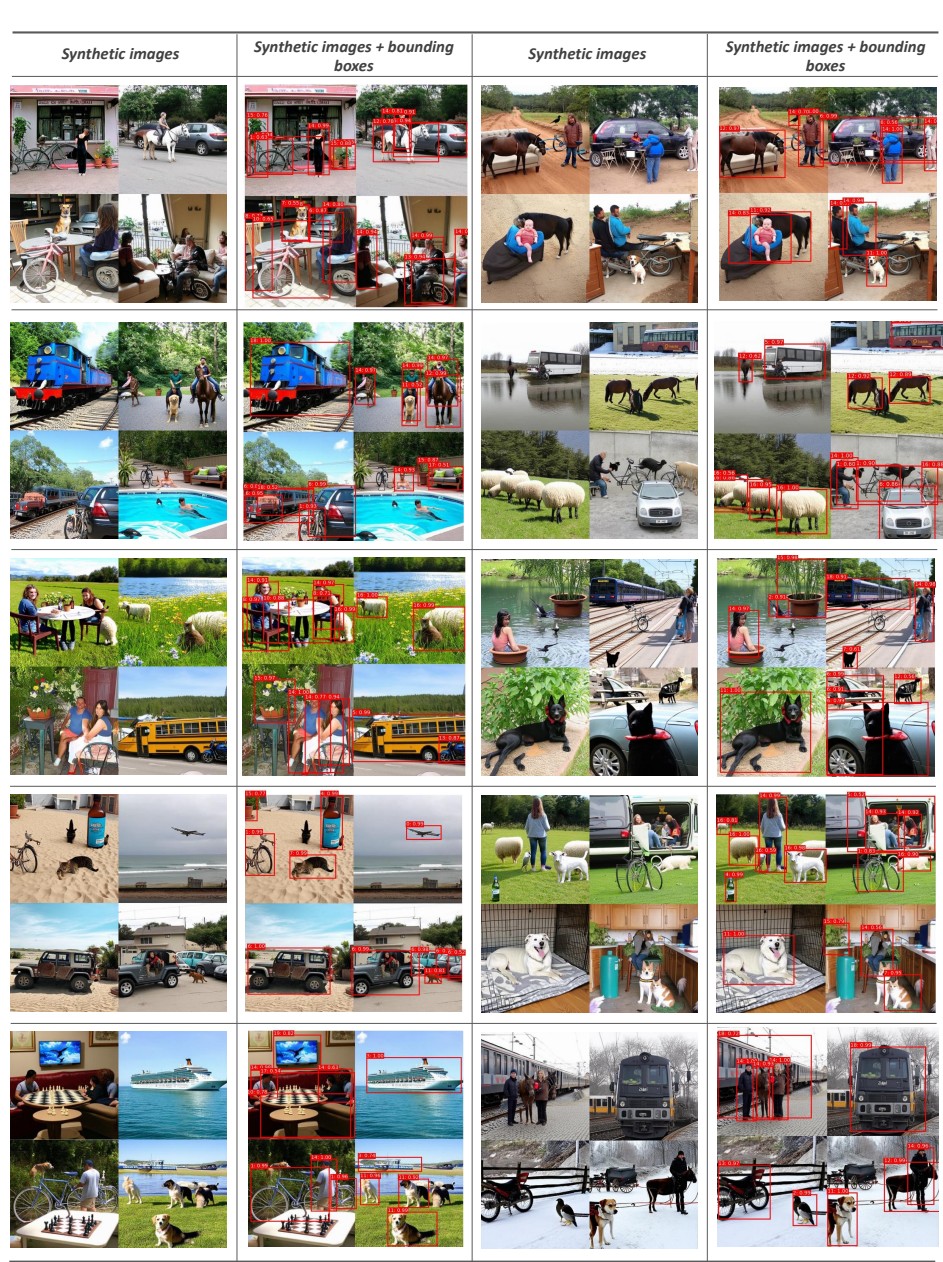

Figure 14: **Generated images on VOC using** $\mathrm{D}^3$**.** Representative samples.

