# OpenReview forum: "$\mathrm{D^3}$: Divide, Describe, and Diffuse: Prompt-Enriched, Scene-Aware Dataset Condensation for Object Detection"
_ICLR.cc/2026/Conference — Submitted to ICLR 2026_

### Official Review · Reviewer_TL4i · 2025-10-31

**Soundness:** 2
**Presentation:** 2
**Contribution:** 2
**Rating:** 4
**Confidence:** 5

**Summary:**

This paper proposes D³ (Divide, Describe, and Diffuse), a prompt-enriched and scene-aware dataset condensation framework for object detection. The method first augments textual prompts through scene-aware caption enrichment, then synthesizes dense and semantically diverse images via a text-to-image diffusion model. Experiments on VOC and COCO demonstrate consistent improvements over recent dataset condensation and coreset-based methods under low-data regimes.

**Strengths:**

1. The proposed framework combines LLM-guided text augmentation and diffusion-based image synthesis in a conceptually clean way.

2. The pipeline is modular, optimization-free, and potentially extensible to other multimodal generation settings.

3. Quantitative results show stable improvements over prior methods (DCOD, UniDD) across multiple detectors and datasets.

**Weaknesses:**

1. Lack of quantitative validation for the “semantic coverage” hypothesis (Sec. 3.1).

The authors claim that detection performance mainly depends on semantic coverage rather than object count. However, this hypothesis lacks quantitative evidence. The observed performance drop could also arise from visual inconsistencies (e.g., illumination, scale mismatch, blending artifacts) rather than semantic sparsity. To substantiate this claim, the authors should:

Design a controlled experiment that fixes the number of pasted objects while varying their semantic diversity, or introduce a CLIP embedding–based diversity metric and analyze its correlation with mAP. Such analysis would convincingly link semantic coverage to detection performance.

2. Weak spatial control compared with existing controllable diffusion models.
The proposed “augment in text space, realize in image space” paradigm is elegant, but its spatial controllability remains weak. Existing controllable diffusion models such as ControlNet, T2I-Adapter, and GLIGEN already provide explicit conditioning via bounding boxes, depth, or pose, ensuring spatial coherence and realistic layouts. In contrast, D³ relies solely on linguistic prompting, which makes object placement and scale less reliable—an issue especially critical for object detection tasks. Incorporating layout-aware conditioning or lightweight structural priors could substantially strengthen the framework.

3. Outdated captioning model and limited linguistic diversity (Sec. 4.1).
The paper uses BLIP for caption extraction, which often yields single-object or coarse scene descriptions. However, modern captioners such as ControlCap (2024) and the Qwen series (e.g., Qwen3-VL) have demonstrated stronger multi-object reasoning and controllable caption generation, often mitigating the very limitations D³ attempts to address via SACE. Thus, the reliance on BLIP introduces unnecessary bias. A more robust design would involve modern multimodal LLMs or controllable captioners (e.g., ControlCap, Qwen-VL) that can directly produce scene-rich, attribute-aware, and region-grounded captions. Additionally, using datasets with human-written captions (e.g., Visual Genome, COCO Captions) could provide a better foundation than automatically generated text.

4. Limited data budget and low absolute performance.

At 0.5% data budget, the model achieves 18.8% mAP on VOC, far below the ≈70% attainable with full data, and only marginally higher than baselines. True dataset condensation should aim to preserve near-full performance with reduced samples, which is not realized here.
The authors should: Extend experiments to ≥5% or 10% budgets, Report mAP–budget curves, and Include multiple runs with standard deviation to establish robustness.

5. Conceptual ambiguity between “dataset condensation” and “synthetic data generation”

The contribution remains conceptually ambiguous. Classical dataset condensation focuses on compressing existing data while preserving task performance;D³, however, generates entirely new synthetic images via text-to-image diffusion, which belongs more to synthetic dataset generation than condensation. Given that the absolute mAP (e.g., 16.4% on COCO vs. ≈40% with full training) remains far below realistic expectations, it is unclear whether the framework achieves meaningful “condensation.” Clarifying this distinction would help the reader properly interpret the paper’s positioning and novelty.

**Questions:**

Please refer to Weakness.

---

> ### Author Response · Authors · 2025-11-20
>
> We are grateful for the reviewer’s comments. Our detailed response is presented below.
>
> ## Response to W1:
>
> We agree with the reviewer that our current experiment does not fully isolate semantic coverage as the sole factor. The main goal of Sec. 3.1 is to show that naïvely increasing raw instance count via pixel-space copy–paste quickly saturates and can even degrade performance due to visual artifacts, occlusion, and context mismatch. These observations motivate our framework, which enriches scenes in text space rather than image space to avoid such pixel-level distortions. We have revised the text accordingly to avoid overclaiming and to clarify the intended scope of the experiment.
>
> ## Response to W2:
>
> We thank the reviewer for the insightful comment. We agree that dedicated controllable diffusion models (e.g., ControlNet, T2I-Adapter, GLIGEN) provide stronger box-level spatial control by conditioning on explicit layouts, depth, or pose. Our setting and goal, however, are slightly different. In $\mathrm{D^3}$, MMC-RA operates by packing multiple captions into a fixed grid and letting the diffusion model “paint” each region according to its caption, thereby offering coarse region-level layout control while substantially enriching semantic content under a strict data budget. Our framework is not designed to compete with ControlNet or GLIGEN in precise box placement; rather, it focuses on increasing scene diversity in a simple, annotation-free manner that is easy to integrate into existing condensation pipelines.
>
> We agree that placing objects at more appropriate locations and scales may further improve performance, and we have added this potential enhancement to the “Discussion, Limitations, and Future Works” section.
>
> ## Response to W3:
>
> |Method|VOC|COCO|
> |-|-|-|
> |ControlCap|8.3$\pm$0.1|4.1$\pm$0.1|
> |Qwen3-VL|8.2$\pm$0.4|4.5$\pm$0.1|
> |BLIP|8.2$\pm$0.1|4.9$\pm$0.2|
> |**BLIP+SACE**|**14.5$\pm$0.2**|**6.8$\pm$0.1**|
>
> We thank the reviewer for the insightful feedback. To address this concern, we conducted additional experiments using two stronger captioners—ControlCap and Qwen3-VL—to replace BLIP during caption extraction. In all cases, the resulting captions were used to synthesize images with the same SD backbone. We further applied our SACE module on top of BLIP captioner.
>
> As shown in the above table, using stronger captioners alone produces performance comparable to BLIP (only a +0.1–0.4 AP difference). In contrast, enriching the BLIP captions with SACE yields a substantially larger improvement (from 8.2 → 14.3 AP on VOC). This indicates that SACE provides complementary benefits beyond what modern captioners offer, and that the gains are not solely due to captioner strength.
>
> ## Response to W4:
>
> Thanks for the valuable comments.
> We also conducted experiments at mid- and low-compression ratios. On the VOC dataset, our method reaches **28.3$\pm$0.3** mAP at 5% of the data, while the random baseline achieves 26.7$\pm$0.2. When increasing above 10%, $\mathrm{D^3}$ remains competitive with the random baseline. Our observations are consistent with those reported in the DCOD[1], where the performance gap between condensed images and random sampling becomes limited beyond 10% of data.
>
> We believe that designing dataset-condensation methods that remain effective at mid- and low-compression ratios is an important future research direction. We have added this point to the “Discussion, Limitations, and Future Works” section in the revised manuscript.
>
> ## Response to W5:
>
> Thanks for the comment. We believe that dataset condensation does not restrict how data should be condensed. Its goal is to improve storage and training efficiency with a limited number of samples. The first dataset-condensation paper synthesized data by directly optimizing the inputs, and since then, many alternative approaches have emerged, including generation-based methods (using diffusion models or GANs) and improved core-set selection strategies. Our method falls into the generation-based category, where we focus on producing densely informative data using VLMs, LLMs, and diffusion models.
>
> Regarding the accuracy of dataset-condensation methods, we agree that although these methods show strong potential for practical applications and are being actively studied, the performance of existing approaches has not yet reached the level required for real-world deployment. We have added this limitation to the new “Discussion, Limitations, and Future Works” section in the revised manuscript.
>
> *References:*
>
> [1] Ding Qi, Fetch and Forge: Efficient Dataset Condensation for Object Detection, NeuRIPS 2025.

---

### Official Review · Reviewer_ZLz9 · 2025-10-31

**Soundness:** 3
**Presentation:** 3
**Contribution:** 3
**Rating:** 6
**Confidence:** 4

**Summary:**

This paper presents D3, a framework for dataset condensation in object detection that leverages text-to-image diffusion models. It addresses the key limitation of prior methods—sparse, single-object scenes—by enriching prompts with scene-aware object lists and using structured composition strategies to generate denser, more diverse synthetic images. The method achieves state-of-the-art results under extreme data constraints on COCO and VOC.

**Strengths:**

The paper presents a well-motivated approach, effectively addressing the core bottleneck of sparse object density in T2I distillation by introducing a scene-to-object dictionary and structured prompting to generate semantically richer, denser synthetic images, leading to significant performance gains under extreme compression.

**Weaknesses:**

1. D³ lacks any explicit mechanism to control or optimize for object scale. Both the scene dictionary and spatial partitioning focus on object presence and rough location, neglecting relative size. This may lead to an unrealistic scale distribution in generated images, potentially under-representing small objects.

2. The "Top-K" frequent objects used in the scene dictionary inherently favor common categories, exacerbating the long-tail problem. The paper offers no strategy (e.g., oversampling rare classes or targeted prompting) to ensure balanced representation of all categories in the distilled dataset.

3. Technically, D³ functions more as an integration of existing models. Its heavy reliance on LLMs (like GPT-4.1) for prompt enrichment introduces significant computational cost and API dependency.

4. The study does not explore performance at higher compression ratios (e.g., 5%, 10%, 20%, 50%), leaving the method's scalability and upper performance bound unclear.

**Questions:**

see Weaknesses

---

> ### Author Response · Authors · 2025-11-20
>
> We thank the reviewer for the valuable feedback. We address the reviewer’s concern below.
>
> ## Response to W1:
>
> We agree that more explicit control over scale distribution (e.g., matching the empirical histogram of box areas or over-sampling very small instances) is an interesting direction. We view this as a complementary extension rather than a fundamental limitation, since our current results indicate that $\mathrm{D^3}$ is already effective even without explicit scale modeling. This is because Stable Diffusion models appear to learn certain physical attributes (e.g., generating objects with appropriate locations, sizes, and relations based on the scene and viewpoint), as reported in [1,2]. Exploring scale-aware and relationship-aware generation is an interesting direction for future work, and we have added this point to the “Discussion, Limitations, and Future Works” section in the revised manuscript.
>
> ## Response to W2:
>
> We appreciate the reviewer’s insightful comment on long-tail categories. We fully agree that considering diversity and data bias in dataset condensation is an interesting research direction, where important data-bias issues (e.g., class imbalance and domain shift) can be incorporated into the condensation process to further improve performance. We have added this research direction to the new “Discussion, Limitations, and Future Directions” section.
>
> Please note that the primary goal and scope of this work is to introduce a linguistic-only scene enrichment framework for dataset condensation, and to investigate whether enriching unused regions of an image with additional semantically coherent objects can increase the training signal per stored sample. To the best of our knowledge, our work is the first linguistic-only scene enrichment framework for dataset condensation, paving the way for new approaches that leverage VLMs, LLMs, and T2I models.
>
> ## Response to W3:
>
> We thank the reviewer for raising this point. We agree that $\mathrm{D^3}$ builds upon pre-trained components such as SD and GPT-based LLMs. However, our core contribution is not the introduction of new architectures, but a novel framework for constructing linguistically enriched, multi-object scenes specifically for dataset condensation. As motivated by the analyses in Section 3, we show that increasing object density is crucial for improving detector performance. $\mathrm{D^3}$ is the first to leverage purely linguistic re-composition to create complex scenes under a fixed storage budget, and this design leads to substantial performance gains over existing baselines.
>
> ## Response to W4:
>
> Thanks for the valuable comments. It is well known that the effective range of dataset condensation has traditionally focused on extremely high compression ratios, as shown in the original dataset-condensation paper [3,4]. Many works have reported that condensed datasets often do not outperform even random selection at mid- and low-compression ratios (e.g., above 10%), as also noted in [4]. This behavior inherently limits the possibility of achieving higher performance than the original dataset at these ratios.
>
> We also conducted experiments at mid- and low-compression ratios. On the VOC dataset, our method reaches **28.3$\pm$0.3** at 5% of the data, while the random baseline achieves 26.7$\pm$0.2. When increasing to 10% and above, $\mathrm{D^3}$ remains competitive with the random baseline. Our observations are consistent with those reported in the DCOD[4], where the performance gap between condensed images and random sampling becomes limited beyond 10% of data.
>
> We believe that designing dataset-condensation methods that remain effective at mid- and low-compression ratios is an important future research direction. We have added this point to the “Discussion, Limitations, and Future Works” section in the revised manuscript.
>
> *References:*
>
> [1] Guanqi Zhan et al., What Does Stable Diffusion Know about the 3D Scene?
>
> [2] Sangwon Baik et al., Learning 3D Object Spatial Relationships from Pre-trained 2D Diffusion Models, ICCV 2025.
>
> [3] Kim et al., Dataset Condensation via Efficient Synthetic-Data Parameterization, ICML 2022.
>
> [4] Ding  Qi et al., Fetch and Forge: Efficient Dataset Condensation for Object Detection, NeuRIPS 2025.

---

### Official Review · Reviewer_sUci · 2025-10-31

**Soundness:** 3
**Presentation:** 3
**Contribution:** 3
**Rating:** 6
**Confidence:** 2

**Summary:**

This paper tackles the problem of dataset condensation (DC) for object detection, identifying a key "representation bottleneck" in prior text-to-image (T2I) methods. The authors argue that naïve conditioning on simple captions produces sparse, single-object images that lack the multi-object co-occurrence and layout diversity essential for training robust detectors. They propose $D^3$ (Divide, Describe, Diffuse), an optimization-free framework that generates dense, semantically diverse synthetic data. The pipeline first Divides the dataset by building a Scene-to-Object Dictionary (SOD) from co-occurrence statistics. It then Describes complex scenes by using an LLM, guided by the SOD, to perform Scene-Aware Caption Enrichment (SACE). Finally, it Diffuses these scenes by using a Multi-Caption Composition (MCC) strategy—specifically a region-aware prompt—to guide a T2I model. The resulting images are pseudo-labeled by a frozen detector. On MS COCO and PASCAL VOC, $D^3$ achieves state-of-the-art results, more than doubling the mAP of the strongest baseline at extremely low data ratios.

**Strengths:**

1. The paper's core strength is its precise diagnosis of why prior T2I-based DC methods fail for detection. The concept of the "representation bottleneck" caused by simple captions is a key insight. This is powerfully supported by the motivating Copy-Paste experiment, which demonstrates that scene coherence and semantic coverage are far more important for detector training than just the raw count of pasted instances.
2. The core idea of "augmenting in text space, then realizing in image space" is a highly effective and novel solution to the problem. The $D^3$ pipeline is clever and logical. In particular, the Scene-to-Object Dictionary (SOD) is a smart mechanism to ground the LLM's creativity in the dataset's actual statistics, ensuring the generated scenes are not just dense, but also realistic.
3. The results are not incremental but represent a massive leap forward, more than doubling the mAP of the previous SOTA (UniDD) in low-data regimes. The high quality of the $D^3$-generated dataset is further validated by its strong generalization: it successfully trains not only the default detector (Faster R-CNN) but also other advanced architectures (DINO, DiffusionDet) and even transfers effectively to other tasks like segmentation and pose estimation.

**Weaknesses:**

1. The framework's knowledge is hard-capped by the capabilities of the frozen pseudo-labeler. The T2I model could generate a perfect object, but if the frozen detector (e.g., Faster R-CNN) fails to recognize it, that object effectively does not exist in the final dataset. This creates a hard ceiling on the condensed data's quality, which can never surpass the knowledge of its labeler.
2. The paper's core claim is that its linguistic layout control (MCC-RA) is a key driver of performance. A critical missing baseline would be a comparison against explicit layout control methods. For instance, sampling realistic bounding box layouts from the dataset and using a layout-to-image model (like GLIGEN) would also capture "multi-object co-occurrence and layout diversity" and would serve as a more direct and challenging baseline.
3. The "optimization-free" claim is misleading, as the framework is a complex, feed-forward cascade of numerous, large-scale, pre-trained models. It requires a VLM (BLIP), a scene classifier (ResNet-50), a large LLM (GPT-4.1), a SOTA T2I generator (SDv3), and a detector (Faster R-CNN). This makes the pipeline computationally expensive to run (approx. 33s per image) and difficult to reproduce, especially for those without API access or massive compute resources.

**Questions:**

None

---

> ### Author Response · Authors · 2025-11-20
>
> Thank you for the helpful feedback. We respond to the reviewer’s concern as follows.
>
> ## Response to W1:
>
> We agree with the reviewer that the quality of condensed data is influenced by the pseudo-labeler. This is an inherent characteristic of all pseudo-label–based dataset-condensation pipelines [1,2]. Our goal in this work, however, is not to surpass the labeler itself, but to automate data condensation under a fixed labeling budget using a reproducible labeling model. Exploring better pseudo-labeling algorithms is an interesting research direction, but it falls outside the scope of our focus on automatic dataset condensation. We have added a “Discussion, Limitations, and Future Works” section noting that improving pseudo-label quality will be an important direction for future research.
>
> ## Response to  W2:
>
> We thank the reviewer for the helpful suggestion. We would like to clarify that GLIGEN plays the same role as Stable Diffusion in our pipeline—both are image generators. In $\mathrm{D^3}$, the core contribution is the linguistic composition of multiple captions (MCC-RA) to create a richer prompt. After this step, SD is simply used to render the composed prompt.
>
> If we wished to use GLIGEN, the pipeline would remain the same:
> $\mathrm{D^3}$ → composed boxes → GLIGEN for rendering.
> In other words, GLIGEN would replace SD only as the renderer, not as a different baseline for dataset condensation. The key idea of $\mathrm{D^3}$ lies in how captions are enriched and composed, not in which T2I model or layout to image is used to generate the image.
>
> Thus, GLIGEN should be viewed as an interchangeable generation module rather than a direct baseline.
>
> ## Response to W3:
>
> We agree that the current method uses several pre-trained models. It is worth noting that modern deep-learning research heavily relies on pre-trained components, and the term “optimization-free” is often used when those models are reused without additional training. In this paper, we adopted “optimization-free” in the context of dataset condensation to distinguish our approach from conventional methods that synthesize condensed images through heavy optimization. However, we agree that the term may be misleading, and we have therefore replaced it with “light-weight” in the revised manuscript. Our main contribution lies in the automatic generation of condensed images, and we believe this modification does not diminish the contribution or impact of our work.
>
> *References:*
>
> [1] Yin, Zeyuan et al., Squeeze, Recover and Relabel: Dataset Condensation at ImageNet Scale From A New Perspective, NeurIPS 2023.
>
> [2] Sung, Pen et al., On the Diversity and Realism of Distilled Dataset: An Efficient Dataset Distillation Paradigm, CVPR 2024.

---

### Official Review · Reviewer_Cxgn · 2025-11-07

**Soundness:** 3
**Presentation:** 3
**Contribution:** 3
**Rating:** 4
**Confidence:** 5

**Summary:**

Dataset Condensation (DC) for object detection is challenging due to the computational cost of bi-level optimization and the information bottleneck of simple text-to-image (T2I) generation models. The proposed framework, D³ (Divide, Describe, Diffuse), addresses this by using a learned scene-to-object dictionary and a two-stage caption enhancement pipeline (including Scene-Aware Caption Enrichment and Multi-Caption Composition) to generate content-rich, spatially diverse synthetic images. This method achieves state-of-the-art performance, significantly outperforming baselines on PASCAL VOC and MS COCO under extreme data constraints.

**Strengths:**

1. While prior DC methods for detection relied on computationally prohibitive bi-level optimization or naïve Text-to-Image (T2I) generation, this work introduces a novel generative paradigm focused on enriching the conditioning signal (captions) rather than heavily optimizing the synthetic images or networks.

2. The results are compelling, achieving state-of-the-art performance under extreme data constraints (0.25% storage budget). The claim of "more than doubling the strongest baselines" on both PASCAL VOC and MS COCO provides strong quantitative evidence of the method's effectiveness and superiority over existing methods like DCOD and UniDD.

**Weaknesses:**

1. The authors state in the introduction that D3 achieves more than twice the improvement compared to UniDD. However, it is important to note that UniDD uses SDv1.5 as its baseline, whereas D3 is built upon SDv3.0. Therefore, this comparison is not entirely fair.
2. The annotations corresponding to the images generated by the authors are derived from the pre-trained detection model Fast R-CNN, which inherently limits their accuracy. Moreover, Faster R-CNN does not represent the state-of-the-art in object detection performance.

**Questions:**

1. In my view, the images ultimately generated by the authors, as shown in Figure 5, bear a strong resemblance to the Mosaic technique—a common data augmentation method used in YOLO series. Mosaic combines four distinct images into a single composite, allowing the model to perceive targets across varying scales and backgrounds. This observation raises an intriguing question: Does the performance improvement reported by the authors’ method primarily stem from the effects of data augmentation?
2. The experiments presented by the authors in Section 3 are quite interesting. However, as noted earlier, the proposed approach bears a strong resemblance to Mosaic. It would be beneficial if the authors could supplement this section with experimental comparisons or results from Mosaic to provide a more comprehensive evaluation.

---

> ### Author Response · Authors · 2025-11-20
>
> We appreciate the reviewer’s helpful feedback. Our response to the reviewer’s concern is provided below.
>
> ##  Response to W1:
>
> |Method|mAP|
> |------------|-------|
> |UniDD|8.5$\pm$0.4|
> |$\mathrm{D^3}$ with SDv1.5|11.4$\pm$0.2|
> |$\mathrm{D^3}$ with SDv3|18.8$\pm$0.1|
>
>
> We thank the reviewer for the constructive comment. To address this point, we additionally performed $\mathrm{D^3}$ using SD-v1.5 on VOC dataset at 0.5% of data as suggested. As shown in the above table, our framework achieves an improvement of approximately 3% mAP over UniDD, indicating that the proposed components provide benefits beyond the choice of backbone. Since our input prompts are more complex than the base captions, a stronger text encoder naturally helps produce more informative conditioning features. This explains why using SD-v3 yields larger gains. We have clarified the reason for choosing SD-v3 in the revised manuscript.
>
> ## Response to W2:
>
> We appreciate the comment. We reported the performance using various detectors for pseudo-labeling and target models in Table 8. While Faster R-CNN is not state-of-the-art in object detection, it offers a stable and reproducible pseudo-labeling setup for dataset-condensation research. We agree that better pseudo-labeling methods could further improve performance, which is an interesting direction. We have included this point in the “Discussion, Limitations, and Future Work” section.
>
> ## Response to Q1:
>
> We appreciate the reviewer’s thoughtful observation. While the synthesized images in Fig. 5 may appear visually similar to Mosaic-style compositions, we do not use mosaic-like images as input to the network. Instead, we randomly crop a sub-region (corresponding to each sub-caption) and feed it to the network. We describe the differences in the table below.
>
> |Aspect|Mosaic|Ours|
> |--|--|--|
> |When to use|Training time|Condensation time|
> |How to use|Entire 4-in-1 image as one sample|Each quadrant used as an independent sample via random cropping|
> |Purpose|Augment the training data for regularization|Condensing more information to limited space|
>
> ## Response to Q2:
>
> Thanks for the comment. As mentioned in the previous question, our method is fundamentally different from Mosaic in terms of purpose, mechanism, and how it is used for training. In Section 3, we showed that mixed objects without overlapping in the dataset-condensation scenario tend to yield higher mAP performance compared to overlapping ones. This motivated us to propose a new framework that uses text manipulation and a T2I generative model to create images with dense yet non-overlapping objects.
> In fact, our region-aware composition is inspired by the reparameterization idea in conventional dataset-condensation methods, where spatial-resolution redundancies [1] are leveraged to encode more information into each synthetic sample.
>
> *References:*
>
> [1] Kim et al., Dataset Condensation via Efficient Synthetic-Data Parameterization (ICML 2022).

---

> > ### Comment · Reviewer_Cxgn · 2025-11-26
> > **Further Clarification Regarding Training Mechanism and Baselines**
> >
> > I thank the authors for their detailed response and the comparison table provided. However, the explanation regarding the "random cropping of sub-regions" and the motivation derived from "non-overlapping mixed objects" raises two critical concerns that require further justification:
> >
> > **1. Fairness of Comparison regarding Data Budget (The "Effective IPC" Issue)**
> >
> > The authors state that for each synthesized image (containing multiple sub-captions/regions), the network is fed a randomly cropped sub-region rather than the full image.In the context of Dataset Condensation, strict adherence to the pixel budget (Images Per Class, IPC) is paramount for fair comparison.Concern: If one synthesized image contains $N$ distinct, non-overlapping sub-regions (e.g., $N=4$ in a quadrant layout), and the model is trained on crops of these regions, are you not effectively increasing the diversity of the training set by a factor of $N$?
> >
> > 2. **Justification for the Grid Size ($N=4$)**
> >
> > The authors default to combining 4 sub-regions. It creates a natural question: Is $N=4$ the optimal trade-off between information density and sample fidelity?
> >
> > + Comparison with $N=2$: Traditional Mixup typically involves 2 samples. Why not use a simple split-screen layout (vertical or horizontal) with 2 sub-regions? Does $N=4$ provide significantly better condensation efficiency than $N=2$?
> >
> > + Scalability to $N>4$: If the goal is to condense more information into limited space, have the authors explored higher grid densities, such as $3 \times 3$ ($N=9$)?

---

> > > ### Author Response · Authors · 2025-11-28
> > >
> > > We thank the reviewer for the prompt feedback. Our response to the additional concerns are provided below.
> > >
> > > ## Concern 1: Fairness of comparison regarding data budget
> > > We appreciate the reviewer’s observation. We agree that placing multiple independent sub-regions inside a single synthesized canvas effectively increases the semantic diversity per stored image. However, this does not violate the IPC constraint, because IPC in DC is defined by stored pixels, not by the number of semantic regions within an image. This principle is widely used and accepted in prior works, such as IDC[1] and RDED[2], where four independently images are downscaled and packed into a single larger canvas to exploit spatial redundancy while still counting as 1 IPC.
> > >
> > > In evaluation, we apply random–region cropping to:
> > > - guarantee that each synthesized image contributes an equal number of training iterations
> > > - avoid unintended mosaic-augmentation effects
> > >
> > > Thereby maintaining fairness in comparison.
> > > Our novelty lies in enriching captions through SACE and generating multi-region images directly in a single SD inference call, achieving the same spatial-efficiency benefits at 4× lower cost.
> > >
> > > ## Concern 2: Justification for the Grid Size
> > >
> > > We thank the reviewer for asking about the choice of grid size. Our use of four sub-regions reflects practical constraints of the generative model and the target task.
> > >
> > > **1. Why not choose $2$ regions?**
> > >
> > > As suggested, we conducted additional experiments with $N = 2$, which produces two possible cases: a vertical split ($1×2$) and a horizontal split ($2×1$). We applied our framework to both scenarios and reported the results on the VOC dataset at a 0.5% condensation ratio using Faster R-CNN.
> > >
> > > || $N = 1×2$  | $N = 2×1$ | $N = 2×2$ |
> > > |-|--------------------|----------------------|---------|
> > > |mAP| 13.4$\pm$0.2       | 15.2$\pm$0.3         | **18.8**$\pm$**0.1** |
> > >
> > > As shown in the table above, both variants of $N = 2$ ($1×2$ and $2×1$) yield substantially lower performance than $N = 4$ under the same training conditions. Therefore, we use $N = 4$ as the default setting.
> > >
> > > **2. Why not choose more than $4$ regions (e.g., $9$ or $16$)?**
> > >
> > > We performed additional experiments with increased $N = 3×3$ and $N = 4×4$. We conducted the experiments on VOC at 0.5% of the data and evaluated with Faster R-CNN.
> > >
> > > || $N = 2×2$            | $N = 3×3$             | $N = 4×4$ |
> > > |-|--------------------|---------------------|---------|
> > > |mAP| **18.8**$\pm$**0.1** | 9.8$\pm$0.4         | 5.3$\pm$0.4 |
> > >
> > > As shown in the table above, using more sub-regions decreases the performance. There are two possible reasons for this.  First, caption length constraints: including 9–16 sub-captions exceeds the maximum token capacity of the text encoder, causing truncation and degrading the quality of the conditioning signal.  Second, resolution constraints: with $9$ or more regions, each sub-image becomes approximately 170×170 pixels or smaller, which is insufficient for SD to generate fine object details—especially small objects that are crucial for detection.
> > >
> > > Due to these reasons, we default to using $4$ regions as the most practical and stable configuration.
> > >
> > > ---
> > > *Reference:*
> > >
> > > [1] Kim et al., Dataset Condensation via Efficient Synthetic-Data Parameterization (ICML 2022).
> > >
> > > [2] Sung, Pen et al., On the Diversity and Realism of Distilled Dataset: An Efficient Dataset Distillation Paradigm, CVPR 2024.

---

### Author Response · Authors · 2025-11-20
**Summary of changes**

We sincerely thank all reviewers for their thorough evaluations and valuable comments. We have made the following revisions (highlighted in blue):

- Minor revisions in Section 3.1 to accurately reflect the intended goal and claim
- In Appendix A.5, the Limitations section has been updated to DISCUSSION, LIMITATIONS, AND FUTURE WORK. All potential improvements raised by the reviewers have been integrated into this section.

We are grateful for the comments that have greatly improved the quality of the manuscript. We hope our updates and explanations adequately respond to your comments.

---

### Author Response · Authors · 2025-12-03
**Summary of Rebuttal, Experiments, and Clarifications**

We thank all reviewers for their constructive and insightful feedback. Due to an unexpected security incident, the discussion period was shortened; below we provide a consolidated summary of our rebuttal. In response to the comments, we conducted several experiments and added substantial clarifications.

---

## Additional Experiments Added:
1. Using Stable Diffusion V1.5[`Cxgn`]: Our framework surpasses UniDD by 2.9% mAP on VOC when using SDv1.5. With SDv3, the improvement becomes more pronounced, achieving a 10.3% mAP gain.
2. Grid-size study[`Cxgn`]: We performed experiments comparing grid layouts of 1×2, 2×1, 2×2, 3×3, and 4×4. Empirically using a grid size of 2x2 achieves the best performance.
3. Stronger captioners (ControlCap, Qwen3-VL)[`TL4i`]: Experiments with stronger captioners (ControlCap, Qwen3-VL) show similar performance, while SACE provides substantially larger improvements, indicating its effectiveness in providing more training signal.
4. Mid- and low-compression ratios[`ZLz9`,`TL4i`]: We conducted experiments under mid- and low-compression ratios and found trends consistent with prior DC literature, and improving performance beyond 10% budget is highlighted as future work.

## Key Clarifications:
1. Different between Mosaic augmentation and our framework[`Cxgn`]: Mosaic modifies pixel layouts and introduces artifacts, whereas D³ enriches scenes linguistically to avoid such distortions.
2. Fairess under IPC setting[`Cxgn`]: We explained how we ensure fair comparison in low-IPC regimes and matched training budgets across baselines.
3. GLIGEN as renderer, not baseline[`sUci`]: GLIGEN would simply replace SD as a rendering module; the core novelty of $\mathrm{D^3}$ lies in linguistic multi-caption composition, not in the choice of T2I model.
4.  “Optimization-free” terminology[`sUci`]: We revised the term to “light-weight” to avoid confusion, as our method uses pretrained models but no gradient-based optimization like conventional DC.

## Future Directions (added to Discussion, Limitation, and Future work section):
1. Improving pseudo-labeling quality[`Cxgn`,`sUci`].
2. Explicit control the object scale[`ZLz9`].
3. Addressing long-tail/class imbalance issue[`ZLz9`].
4. Explicit box-controlling with GLIGEN/ControlNet[`sUci`,`TL4i`].
5. Design DC that is effective for mid and low compression ratios.[`ZLz9`,`TL4i`]

---

Overall, we strengthened the paper with new empirical studies, clarified the scope and novelty of $\mathrm{D^3}$, and outlined concrete future directions, while demonstrating that D³ consistently improves dataset condensation through linguistic scene enrichment.

---

### Meta-Review · Area_Chair_4k28 · 2026-01-10

**Summary:**

The reviewers express different opinions on this submission, which are summarized as in the following.

1. The technical novelties are limited [Review ZLz9].
2. Limited evaluations. The paper targets at one new pipeline for DC, as such sufficient evaluation is extremely important. The reviewers expresses some concerns on the evaluation, quantiative validation for "semantic coverage" [reviewer TL4i], week spatial control [reviewer TL4i], missing baseline with layout control.
3. The limitations of the proposed method. The object scale problem is not considered [Reviewer ZLz9],  the long-tail problem [Reviewer ZLz9], hard ceiling of its labeler [Reviewer sUci],

In my opinion, the paper propose one systematical pipeline for DC, using existing models. As such, the motivation, strengths, shortcomings of the proposed method needs to be fully considered. However, in this paper, several key components, which are also agreed with the authors, such improving pseudo-labeling quality, explicit control of the object scale, long-tail imbalance issue, are not considered, which are leaving for the future work.

As I think that the paper is not ready for publication, which should consider and address these problems.

**Reviewer Concerns:**

The limited evaluations of the proposed method have been clarified.
The limitations are still outstanding, which are leaved as future work.

**Reviewer Scores:**

I think that the authors will maintain their previous ratings.

---

### Decision · Program_Chairs · 2026-01-26

Reject